# Plasmon-driven synthesis of individual metal@semiconductor core@shell nanoparticles

Rifat Kamarudheen[1,2], Gayatri Kumari [1,2] & Andrea Baldi[1,2,3 ✉]

Most syntheses of advanced materials require accurate control of the operating temperature. Plasmon resonances in metal nanoparticles generate nanoscale temperature gradients at their surface that can be exploited to control the growth of functional nanomaterials, including bimetallic and core@shell particles. However, in typical ensemble plasmonic experiments these local gradients vanish due to collective heating effects. Here, we demonstrate how localized plasmonic photothermal effects can generate spatially confined nanoreactors by activating, controlling, and spectroscopically following the growth of individual metal@semiconductor core@shell nanoparticles. By tailoring the illumination geometry and the surrounding chemical environment, we demonstrate the conformal growth of semiconducting shells of $CeO_2$, ZnO, and ZnS, around plasmonic nanoparticles of different morphologies. The shell growth rate scales with the nanoparticle temperature and the process is followed in situ via the inelastic light scattering of the growing nanoparticle. Plasmonic control of chemical reactions can lead to the synthesis of functional nanomaterials otherwise inaccessible with classical colloidal methods, with potential applications in nanolithography, catalysis, energy conversion, and photonic devices.

[1] DIFFER - Dutch Institute for Fundamental Energy Research, De Zaale 20, 5612 AJ Eindhoven, The Netherlands. [2] Institute for Complex Molecular Systems, Eindhoven University of Technology, P.O. Box 513, 5600 MB Eindhoven, The Netherlands. [3] Department of Physics and Astronomy, Vrije Universiteit Amsterdam, De Boelelaan 1081, 1081 HV Amsterdam, The Netherlands. ✉email: a.baldi@vu.nl

Accurate control over the operating temperature, pressure, and reactant concentration is quintessential for the synthesis of advanced materials with tailored functionalities[1–3]. Especially in nanoparticle syntheses, the temperature of the reaction mixture often dictates their structural features such as crystallinity, size, shape, porosity, and surface area[4]. Rigorous control over the nanoparticle morphology is fundamental in engineering their optical, electronic, and catalytic properties. For example, different morphologies of ceria nanoparticles such as rods, cubes, and spheres can be synthesized by tuning the reaction temperature, where each shape exhibits a different catalytic activity to CO oxidation[5]. Along with conventional heating, temperature gradients can also be used to manipulate the self-assembly of nanoparticles, such as for the formation of nematic liquid crystalline orders[6]. Another strategy to control the morphology of nanostructures is by driving their synthesis in spatially confined volumes, typically known as nanoreactors[7]. Plasmonic heating in metal nanostructures can generate such nanoreactors, by activating local temperature gradients at their surface to drive the synthesis of advanced functional materials.

Localized surface plasmon resonances (LSPRs) are collective oscillations of free electrons in metal nanostructures[8,9], leading to strong light absorption and scattering[10]. Under continuous wave (cw) excitation of a spherical particle, these resonances generate localized temperature increases, $\Delta T$, at the surface according to[11]:

$$\Delta T = \frac{\sigma_{abs}I}{4\pi\kappa R} \qquad (1)$$

where $\sigma_{abs}$ is the absorption cross-section of the nanoparticle, $I$ is the irradiance (optical power per unit area), $\kappa$ is the thermal conductivity of the surrounding medium, and $R$ is the nanoparticle radius. For example, for a gold nanosphere of 50 nm in radius immersed in water ($\kappa_{H_2O}(20\,°C) = 0.6$ W/m/K) and illuminated with a 532-nm cw laser ($\sigma_{abs}(532\,nm) = 2.08 \times 10^{-14}$ m$^2$) with an intensity of 1 mW/μm$^2$, the temperature at the surface of the nanoparticle will be 55 K higher than in the surrounding medium. Such a large temperature increase, which decays as the inverse of the distance from the nanoparticle surface[12], can significantly accelerate temperature-dependent reactions according to the Arrhenius equation[13]. The use of these localized temperature gradients can circumvent spatial anisotropies encountered in classical colloidal nanomaterial syntheses, for instance due to the different reactivity of crystalline facets or due to the presence of multiple nucleation sites. However, in ensemble photochemical experiments, where the illumination area is typically much larger than the average interparticle distance, collective photothermal effects often dominate over the localized temperature gradients[14–17]. For this reason, the synthesis of well-defined nanomaterials using plasmonic nanoreactors requires the resonant excitation of individual nanoparticles.

Previously, localized-photothermal heating has been used to synthesize a range of nanomaterials, such as carbon nanotubes[18], germanium nanowires[19], and gold nanoellipsoids[20]. In these studies, collective photothermal heating, electric and magnetic dipole resonances, and hot-charge carriers additionally contributed to nanoparticle growth, leading to non-trivial morphologies of the final products. More recently, a combination of localized and collective photothermal effects have been employed to drive the growth of inorganic crystals on plasmonic structures of a few hundreds of nanometers in size[21]. However, so far it has been difficult to use nanoscale temperature gradients selectively in plasmonic nanoparticles to drive the synthesis of isotropic semiconducting shells with tunable chemical compositions and controlled thicknesses down to a few nanometers.

In this work, we present a versatile technique to synthesize, optically control, and in situ track the growth of metal@semiconductor core@shell nanostructures using localized plasmonic heating. We achieve this goal by exciting the plasmon resonance of individual colloidal Au nanoparticles in a tailored reactive chemical environment, and by simultaneously following the reaction in situ and in real time by measuring the photoluminescence (PL) of the growing core@shell nanoparticle. In a typical experiment, a 532-nm cw laser is focused on a single Au nanoparticle to drive the growth of a semiconducting shell using local temperature gradients, without activating any macroscopic collective photothermal effects. Polarization-dependent scattering spectra of the photothermally grown core@shell nanoparticles confirm the isotropic nature of the shell. In addition, we demonstrate that we can control the shell thickness at the nanometer level by optically tuning the nanoparticle surface temperature. The growth of these core@shell nanoparticles can be monitored in situ by tracking the spectral evolution of their PL, which closely resembles their plasmon resonance. We demonstrate the versatility of plasmonic heating as a chemical activation mechanism by synthesizing Au nanoparticles coated with CeO$_2$, ZnO, and ZnS. Strikingly, localized heating effects in individual plasmonic nanoreactors can drive the synthesis of isotropic core@shell nanospheres that are inaccessible under identical chemical conditions in an ensemble reaction. Such unique characteristics of localized-photothermal growth are also used to demonstrate patterning of a two-dimensional substrate with semiconductors of different chemical compositions.

## Results

**Photothermal synthesis: proof of concept.** A purely temperature-driven synthesis is selected to demonstrate the photothermal activation of chemical reactions on Au nanoparticles. For this purpose, we choose a synthesis of Au@CeO$_2$ core@shell nanoparticles that leads to a CeO$_2$ shell thickness of ~10 nm over 1 h heating at 90 °C, but is too slow at room temperature to lead to any significant shell growth[22]. In a typical synthesis, a Ce$^{3+}$ salt and ethylenediaminetetraacetic acid (EDTA) are added to gold nanoparticles stabilized by cetyltrimethylammonium bromide (CTAB) in water. At higher temperatures, Ce$^{3+}$-EDTA complexes are more easily hydrolyzed and Ce$^{3+}$ ions are oxidized by the dissolved O$_2$ at the surface of gold nanoparticles, which act as nucleation sites for the formation of CeO$_2$. The shell growth is confirmed spectroscopically by a red shift of the plasmon resonance of the nanoparticles, attributed to the larger refractive index of the shell surrounding the Au spheres (see Methods and Supplementary Note 1).

The reaction protocol used in the ensemble synthesis is then adapted to grow individual Au@CeO$_2$ core@shell nanoparticles using a plasmonic nanoreactor approach (see Methods). Colloidally synthesized Au nanospheres with a diameter of 66 ± 3 nm are used to photothermally grow core@shell nanostructures under cw illumination with a 532-nm laser (Supplementary Note 2)[16]. This nanoparticle size leads to large localized heating effects according to Eq. (1), as it combines a high absorption cross-section at the irradiation wavelength with a small radius (see also Supplementary Note 3). A quartz substrate covered with optically isolated Au nanoparticles is used as the bottom window of a flow cell placed on an inverted microscope. Dark-field scattering spectroscopy is used to characterize the plasmon resonance of individual nanoparticles before and after the photothermal growth of a ceria shell (Fig. 1a)[23]. Single Au nanoparticles are discriminated from nanoparticle clusters by comparing the measured scattering spectra of individual bright spots in dark-field mode with the one expected from a single 66-nm Au nanoparticle. A single Au nanoparticle is then irradiated for 15 min with a 532-nm cw laser,

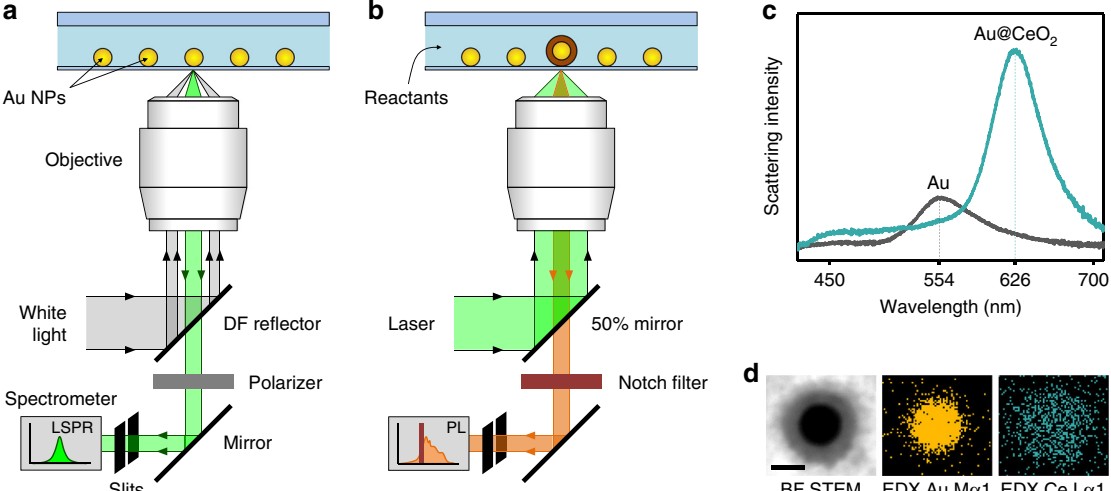

**Fig. 1 Photothermal growth of core@shell nanostructures.** Schematic illustration of the inverted dark-field microscope configuration (**a**) when used to probe the scattering spectra of individual plasmonic nanoparticles under white light illumination and (**b**) when used for 532 nm continuous wave laser irradiation and for simultaneously measuring the photoluminescence spectra of the growing core@shell nanoparticle. **c** Dark-field scattering spectra of a single Au nanoparticle (gray) and of the corresponding photothermally grown Au@CeO$_2$ core@shell nanoparticle (teal) on a quartz substrate, after 15 min of irradiation. **d** BF-STEM image and corresponding Au Mα1 (yellow) and Ce Lα1 (teal) EDX maps of an Au@CeO$_2$ core@shell nanoparticle photothermally grown on a silicon nitride membrane. The scale bar corresponds to 50 nm.

while flowing a solution of Ce$^{3+}$ ions and EDTA (see Fig. 1b and details in Methods). The irradiation intensity is tuned between 0 and 19 mW/μm$^2$ to generate an estimated increase in the nanoparticle surface temperature of up to 500 K (Supplementary Note 4). During laser irradiation, we continuously measure the PL spectra of the growing core@shell particle. After laser irradiation, we switch back to white light illumination and measure the dark-field scattering spectrum of the final Au@CeO$_2$ nanoparticle.

In Fig. 1c, we plot the dark-field scattering spectra of a gold nanoparticle before and after laser irradiation with a power density of 11 mW/μm$^2$ for 15 minutes. The CeO$_2$ shell growth is visible as a color change in the dark-field image of the nanoparticle from green to orange (Supplementary Note 5). A red shift of ~70 nm is observed in the plasmon resonance of the irradiated nanoparticle, along with an increase in the scattering intensity and a narrowing of its full width at half maximum (FWHM) by ~50 meV (Fig. 1c, teal curve). These spectral changes are consistent with the growth of a ceria shell of ~15 nm in thickness around the irradiated Au nanosphere (Supplementary Note 6). With our experimental procedure we can determine the plasmon resonance position of our nanoparticles with a typical accuracy of ±0.01 eV, which corresponds to a ceria shell thickness uncertainty of less than 1 nm (Supplementary Note 6).

To confirm the photothermal growth of CeO$_2$ upon laser irradiation, we performed an additional experiment using Au nanoparticles deposited on a silicon nitride transmission electron microscope (TEM) membrane (see details in the "Methods" section). Figure 1d shows a bright-field scanning transmission electron microscopy (BF-STEM) image and the corresponding energy dispersive X-ray (EDX) maps for one of the photothermally grown Au@CeO$_2$ nanoparticles. No CeO$_2$ shell growth is observed for nanoparticles that are on the same TEM membrane but are not irradiated, in agreement with our optical scattering studies (Supplementary Note 7).

In principle, correlated optical-electron microscopy could be used to determine the final shell thickness of our plasmonically

grown Au@CeO$_2$ nanoparticles. However, the use of ultrathin, fragile TEM membranes in liquid flow cells is experimentally challenging and cannot be relied upon to obtain significant statistics. We therefore base our shell thickness derivation upon the measured spectral shifts in the irradiated nanoparticles using dark-field scattering spectroscopy.

As an additional control experiment, we also measured the dark-field spectra of Au nanoparticles before and after laser irradiation in the absence of Ce$^{3+}$ ions. The spectra perfectly overlap, indicating that the nanoparticles are stable under our laser irradiation intensities (Supplementary Note 7).

The CeO$_2$ shell growth is an oxidation reaction and could therefore in principle be assisted by photo-generated plasmonic or interband hot holes in the gold nanoparticles[24]. However, hot holes generated in gold under 532 nm illumination are not energetic enough to be injected in the growing CeO$_2$ shell (Supplementary Note 8)[22], suggesting that photothermal effects are the sole contributors to the reaction mechanism.

**Tunable and isotropic shell growth.** In a typical experiment, we focus a Gaussian laser beam of ~2 μm in diameter to excite a single gold nanoparticle. To estimate the increase in the nanoparticle surface temperature, we adopt the following protocol. We first determine the relative position of the nanoparticle with respect to the laser spot in the sample plane by fitting the dark-field scattering image of the nanoparticle (Fig. 2a) and the 532-nm irradiation laser spot (Fig. 2b) using two-dimensional Gaussian functions[25]. These fitting routines give us the relative position of the particle with respect to the center of the laser with a typical in-plane accuracy of ±100 nm. From the measured LSPR of the Au nanoparticle, we extract its radius $R$, and its absorption cross-section $\sigma_{abs}$, using finite-difference time-domain (FDTD) calculations (see Methods) and assuming a perfect spherical shape of the nanoparticle supported on quartz. Using the measured laser power at the sample $P$, the effective thermal conductivity of the surrounding medium $\kappa$ (Supplementary Note 9), the nanoparticle in-plane

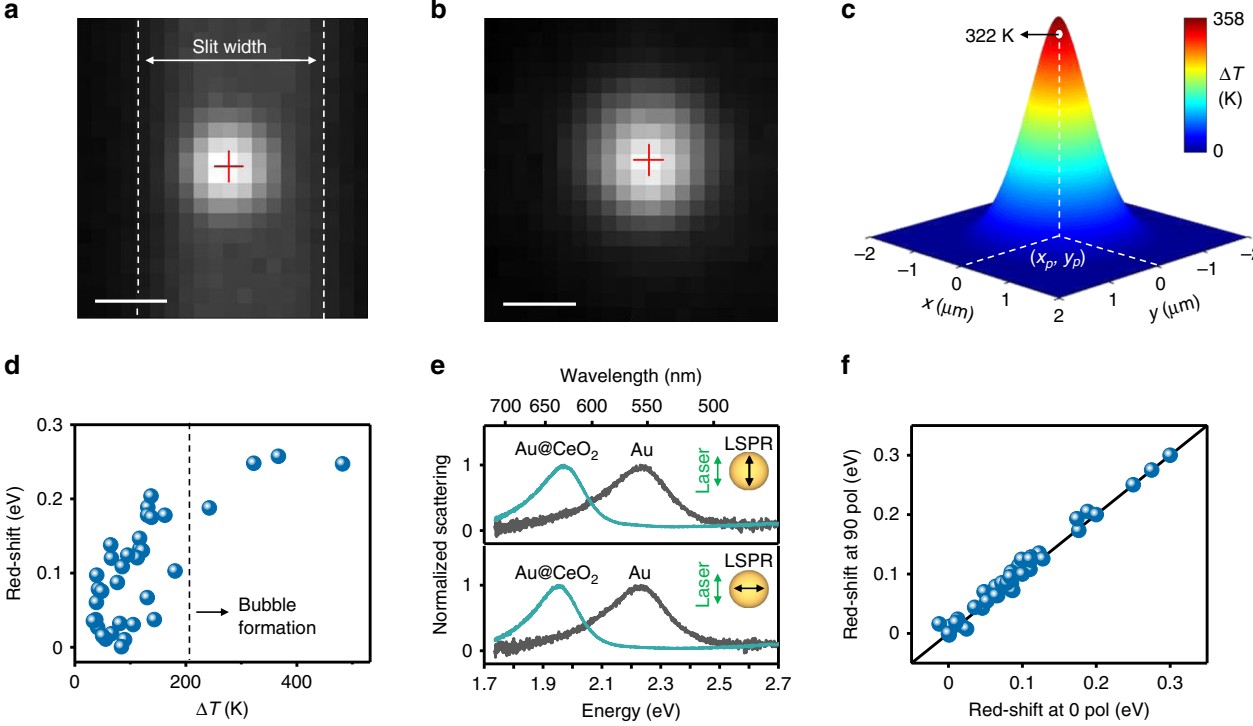

**Fig. 2 Analysis of photothermally grown nanoparticles.** Representative EM-CCD micrographs of **a** a single Au nanoparticle imaged in dark-field mode and **b** the laser beam profile reflected at the substrate/solution interface. Both images are fitted with a 2D Gaussian function and the red crosses mark their fitted center positions. The scale bars in **a** and **b** correspond to 1 μm. **c** Colormap representation of the temperature increase $\Delta T(x, y)$, for a Gaussian laser beam profile with a total optical power of 18 mW and a standard deviation $\sigma_l = 511$ nm. The temperature increase is plotted for a nanoparticle with a dark-field scattering spectrum peak at 554 nm, from which, using FDTD, we calculate the nanoparticle radius (36.5 nm) and absorption cross-section at 532 nm ($1.6 \times 10^4$ nm$^2$). The white dot corresponds to the estimated nanoparticle temperature under laser irradiation, calculated using the fitted in-plane coordinates $(x_p, y_p)$ for the particle and laser positions shown in panels **a** and **b**. **d** Red shift of the scattering spectra of 35 different single Au@CeO$_2$ core@shell nanoparticles after the laser-driven shell growth, plotted as a function of the calculated initial nanoparticle temperature increase. To ensure reproducibility, the data points have been collected over five different sets of experiments using newly prepared flow cell assemblies and reactant solutions. The dashed line at 210 K represents the bubble formation temperature typically observed under nanoparticle photothermal heating[28]. **e** Scattering spectra recorded through a polarizer filter aligned both parallel (top) and perpendicular (bottom) to the laser polarization for a typical Au nanoparticle before (gray) and after (teal) photothermal CeO$_2$ shell formation. The red-shift measured over the two orthogonal polarizations is identical, indicating an isotropic shell growth. **f** Red-shift measured after ceria shell growth for several nanoparticles at orthogonal polarizations. The black line corresponds to a 1:1 correlation.

coordinates $(x_p, y_p)$, and the laser in-plane coordinates and standard deviation $(x, y, \sigma_l)$, we estimate the initial increase in nanoparticle surface temperature (Fig. 2c) according to[26]:

$$\Delta T(x, y) = \frac{\sigma_{abs} P}{8\pi^2 \kappa R \sigma_l^2} \exp\left(-\frac{(x - x_p)^2 + (y - y_p)^2}{2\sigma_l^2}\right) \quad (2)$$

By varying the laser intensity, we can modulate the nanoparticle surface temperature and therefore the final thickness of the grown ceria shell. In Fig. 2d, we plot the measured red shift of the plasmon resonance upon ceria shell growth after laser irradiation as a function of the estimated surface temperature increase for 35 different single-particle experiments. The measured spectral shifts after ceria shell growth roughly scale with the calculated surface temperature. The scatter in the data in Fig. 2d mainly originates from the error in the estimated temperature, due to the small drift in the nanoparticle position during the 15 min of laser irradiation. Interestingly, when the estimated temperature increase is above ~200 K, the photothermal red shift of the plasmon resonance reaches a plateau, indicating that heating above this threshold does not further enhance ceria shell growth. We attribute such an effect to the photothermal formation of

water vapor bubbles around the gold nanoparticles, which has previously been observed at similarly high temperatures[27]. The nanoscale radius of curvature of the Au nanoparticles, in fact, allows superheating of water above its standard boiling point, as large Laplace pressures are required to sustain nanoscopic bubble formation[28]. Our nanoscale plasmonic reactors therefore effectively act as optically driven analogues to hydrothermal bombs, which are often used for the synthesis of high quality, single-crystalline materials. Note that the temperature distribution around the irradiated plasmonic nanoparticle is not distorted by the presence of photothermal convective flows, thanks to the low Rayleigh number of water[28].

We further find that the photothermally grown ceria shells are isotropic in nature, as confirmed by polarization-dependent scattering spectra on several individual nanoparticles (Fig. 2e, f). In our measurements, both Au and Au@CeO$_2$ nanoparticles exhibit similar LSPRs at perpendicular polarizations, confirming their isotropic morphology before and after irradiation. Chemical reactions driven by plasmonic hot-charge carriers and near-field enhancements usually display a spatial distribution that depends on the laser polarization[20,29,30]. The isotropic nature of the core@shell nanoparticles observed in Fig. 2f is

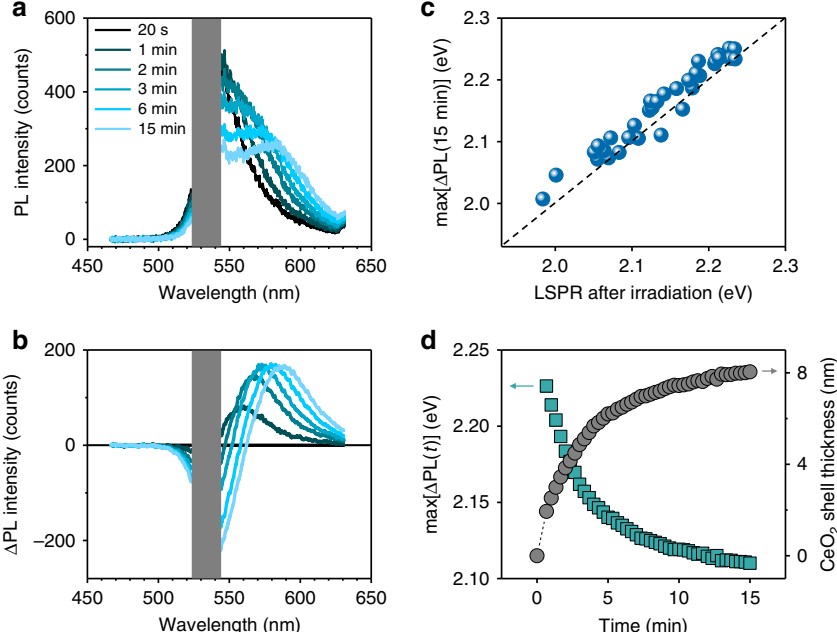

**Fig. 3 Tracking shell growth using photoluminescence. a** Time evolution of the inelastic emission (IE) spectra measured over a period of 15 min, under 532-nm irradiation during the photothermal growth of a CeO₂ shell around an Au nanoparticle deposited on quartz. **b** Time evolution of the ΔPL spectra obtained by subtracting all the IE spectra at time *t*, with the first IE spectrum measured at *t* = 20 s. The gray areas in **a** and **b** correspond to the spectral rejection bandwidth of the 532 nm notch filter used to prevent laser damage of the spectrometer. **c** Maximum of the ΔPL spectra at *t* = 15 min, max[ΔPL (15 min)], versus the LSPR after irradiation, for multiple photothermally grown Au@CeO₂ core@shell nanoparticles. The black dashed line corresponds to a 1:1 correlation. **d** Time evolution of (teal squares, left *y*-axis) the maximum of the ΔPL spectra, max[ΔPL(*t*)], extracted from **b** and (gray circles, right *y*-axis) the corresponding estimated thickness of the growing ceria shell.

therefore a further confirmation that the shell growth is guided by the uniform nanoparticle surface temperature[13].

**In situ tracking of core@shell nanoparticle growth**. Under 532-nm irradiation, the growth of a dielectric shell around Au nanoparticles results in a red shift of their plasmon resonance and in a decrease of their absorption cross-section at the illumination wavelength (Supplementary Note 10). These spectral shifts lead to a lower nanoparticle surface temperature and a subsequent slowing down of the shell growth over the course of the light-driven synthesis. To track the photothermal growth kinetics in situ, we follow the inelastic Stokes emission, or PL, of the nanoparticles during laser irradiation (Fig. 1b). While the origin of PL emission in metals is still under debate[31,32], it is commonly accepted that the PL spectral shape for small Au nanoparticles closely follows their LSPR[33]. By tracking the PL emission, a wealth of information about the photothermally growing nanoparticle can be extracted, such as the evolution of its plasmon resonance, the shell thickness, and the nanoparticle surface temperature. In Fig. 3a, we plot a representative time evolution of the inelastic emission (IE) signal recorded during the photothermal growth of a CeO₂ shell on a single Au nanoparticle (see Supplementary Note 11 for the complete dataset). The measured inelastic scattering spectrum is a superposition of different contributions, including the PL from the growing core@shell nanoparticle, the luminescence of the underlying quartz substrate, and the surface-enhanced Raman scattering (SERS) of reactants on the nanoparticle surface. In our experiments we don't see evidence of any SERS contribution, most likely due to the weak near-field enhancement of our gold nanoparticles. Assuming negligible SERS signal, we isolate the nanoparticle PL from the other contributions by subtracting

all the IE spectra recorded over the course of 15 min of illumination with the first measured IE spectrum, according to (see also discussion in Supplementary Note 11):

$$
\begin{aligned}
\Delta\mathrm{PL}(t) &= \mathrm{IE}(t) - \mathrm{IE}(t=0) \\
&= \left[\mathrm{PL}_{\mathrm{Au@CeO_2}}(t) + \mathrm{PL}_{\mathrm{quartz}} + \mathrm{SERS}\right] - \left[\mathrm{PL}_{\mathrm{Au}} + \mathrm{PL}_{\mathrm{quartz}} + \mathrm{SERS}\right] \\
&= \mathrm{PL}_{\mathrm{Au@CeO_2}}(t) - \mathrm{PL}_{\mathrm{Au}}
\end{aligned}
$$

$$(3)$$

Since the PL of gold nanoparticles, in first approximation, is proportional to their scattering cross-section, $\sigma_{\mathrm{sca}}$[32–34], we can therefore write:

$$
\Delta\mathrm{PL}(t) \propto \left[\sigma_{\mathrm{sca,Au@CeO_2}}(t) - \sigma_{\mathrm{sca,Au}}\right] \quad (4)
$$

For sufficiently large shell thicknesses, for which only a small spectral overlap is expected between the scattering cross-sections of the initial and final particles, the maximum of the subtracted spectrum ΔPL(*t*), henceforth referred as max[ΔPL(*t*)], roughly corresponds to the LSPR of the growing Au@CeO₂ nanoparticle (Fig. 3b and also Supplementary Note 11 for the complete dataset):

$$
\max[\Delta\mathrm{PL}(t)] \approx \max\left[\sigma_{\mathrm{sca,Au@CeO_2}}(t)\right] \quad (5)
$$

An excellent correlation between the max[ΔPL] at *t* = 15 min and the LSPR is in fact observed for several photothermally synthesized Au@CeO₂ core@shell nanoparticles, as shown in Fig. 3c. An average blue-shift of 19 meV is measured for the max [ΔPL] at *t* = 15 min with respect to the corresponding plasmon resonance of the final Au@CeO₂ nanoparticles. This discrepancy can be explained by the high surface temperature of nanoparticles during irradiation, which decreases the refractive index of the

surrounding water and blue-shifts their plasmon resonance[35]. The measured blue-shift of the PL compared to the LSPR of plasmonic nanoparticles may also be due to multiple photon absorptions, which has also been previously reported at laser intensities similar to the ones used in the present study (Supplementary Note 11)[36].

In Fig. 3d, we plot the max[ΔPL] and the corresponding extrapolated ceria shell thickness as a function of time, for the data extracted from Fig. 3b. We observe a 90% shell growth completion in ~10 min for an initial estimated surface temperature increase of ~200 K (Supplementary Note 12). As the thickness of the ceria shell increases, the plasmon resonance of the nanoparticle shifts away from the wavelength of the excitation laser, until no more resonant excitations can occur. This shift leads to a decrease of the shell growth rate with time, showing how, under properly tuned optical illumination conditions, the photothermal growth of metal@semiconductor nanostructures can be conducted as a self-limited process.

**Versatility of photothermal shell growth.** So far, we have limited our discussion to the photothermal growth of ceria shells over plasmonic Au nanospheres. Here, we demonstrate the versatility of our plasmonic nanoreactor approach by photothermally growing different semiconducting shells over plasmonic nanospheres and nanorods. We first demonstrate the photothermal activation and the corresponding in situ PL tracking of a ceria shell growth over an individual Au nanorod. We then photothermally grow various semiconductors on Au nanospheres by varying the cation from $Ce^{3+}$ to $Zn^{2+}$ to grow ZnO, and then by varying the anion from $O^{2-}$ to $S^{2-}$ to form ZnS shells.

Using a 532-nm cw laser, we excite the transverse resonance corresponding to the short axis of an individual Au nanorod (Au NR) deposited on a quartz substrate, to drive the photothermal growth of a $CeO_2$ shell (Fig. 4a and Methods). In Fig. 4b, we plot

the longitudinal LSPR of the Au NR measured before and after irradiation. Due to the narrow width of the nanorod (~25 nm), the transverse plasmon resonance, which is expected at ~500 nm, is not visible in our measured scattering spectra (see Supplementary Note 13). The longitudinal plasmon resonance corresponding to the long axis of the nanorod red-shifts by ~12 nm, indicating the growth of a thin $CeO_2$ shell of ~1 nm in thickness (Supplementary Note 13). This ceria shell is thinner than the one grown on Au nanospheres, because of the lower absorption cross-section of the nanorod at the excitation wavelength and therefore its lower photothermal heating (Supplementary Note 14). Under 532-nm irradiation, the longitudinal plasmon resonance of the Au NR spectrally overlaps with the strong Raman signal from water (Fig. 4c). Nevertheless, the differential photoluminescence signal, ΔPL, is capable of revealing the plasmon resonance of the growing particle (Fig. 4d). Our experiments on nanorods demonstrate that, regardless of their shape, plasmonic nanoparticles can be used to synthesize and in situ track the photothermal growth of core@shell nanoparticles.

In Fig. 5, we show a comparison between the photothermal growth of $CeO_2$, ZnO, and ZnS semiconducting shells over individual Au nanospheres. The synthesis of ZnO shells is driven by the photothermal decomposition of hexamethylenetetramine (HMT) molecules, which leads to a local increase in the pH near the nanoparticle surface to catalyze the hydrolysis of zinc nitrate to zinc oxide (see also details in Methods)[37]. Contrary to the case of Au@$CeO_2$ (Fig. 5d), the scattering spectrum of the photothermally grown Au@ZnO nanosphere (Fig. 5e) exhibits multiple peaks due to geometrical Mie resonances originating from a shell thickness of hundreds of nanometers (Supplementary Note 15). Such a large shell thickness of ZnO compared to the one of $CeO_2$ grown under similar laser intensities can be attributed to the faster reaction kinetics of ZnO formation under our experimental conditions. Indeed, ensemble nanoparticle syntheses performed at

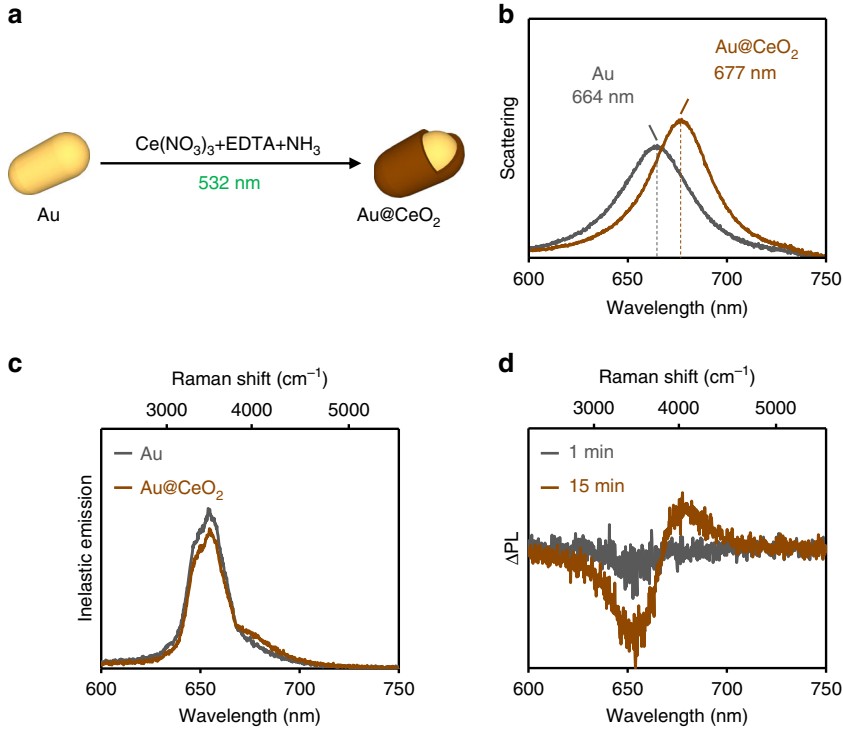

**Fig. 4 Photothermal shell growth on an Au nanorod. a** Schematic representation of the photothermal synthesis of an Au@$CeO_2$ core@shell nanorod (NR). **b** Dark-field scattering spectra of a single Au NR (gray) before and (brown) after the photothermal growth of a $CeO_2$ shell. **c** Inelastic emission spectra collected from the Au NR in **b**, under irradiation using a 532 nm cw laser at $t = 0$ min (gray) and $t = 15$ min (brown). **d** ΔPL spectra of the photothermally grown Au@$CeO_2$ core@shell nanorod at $t = 1$ min (gray) and $t = 15$ min (brown).

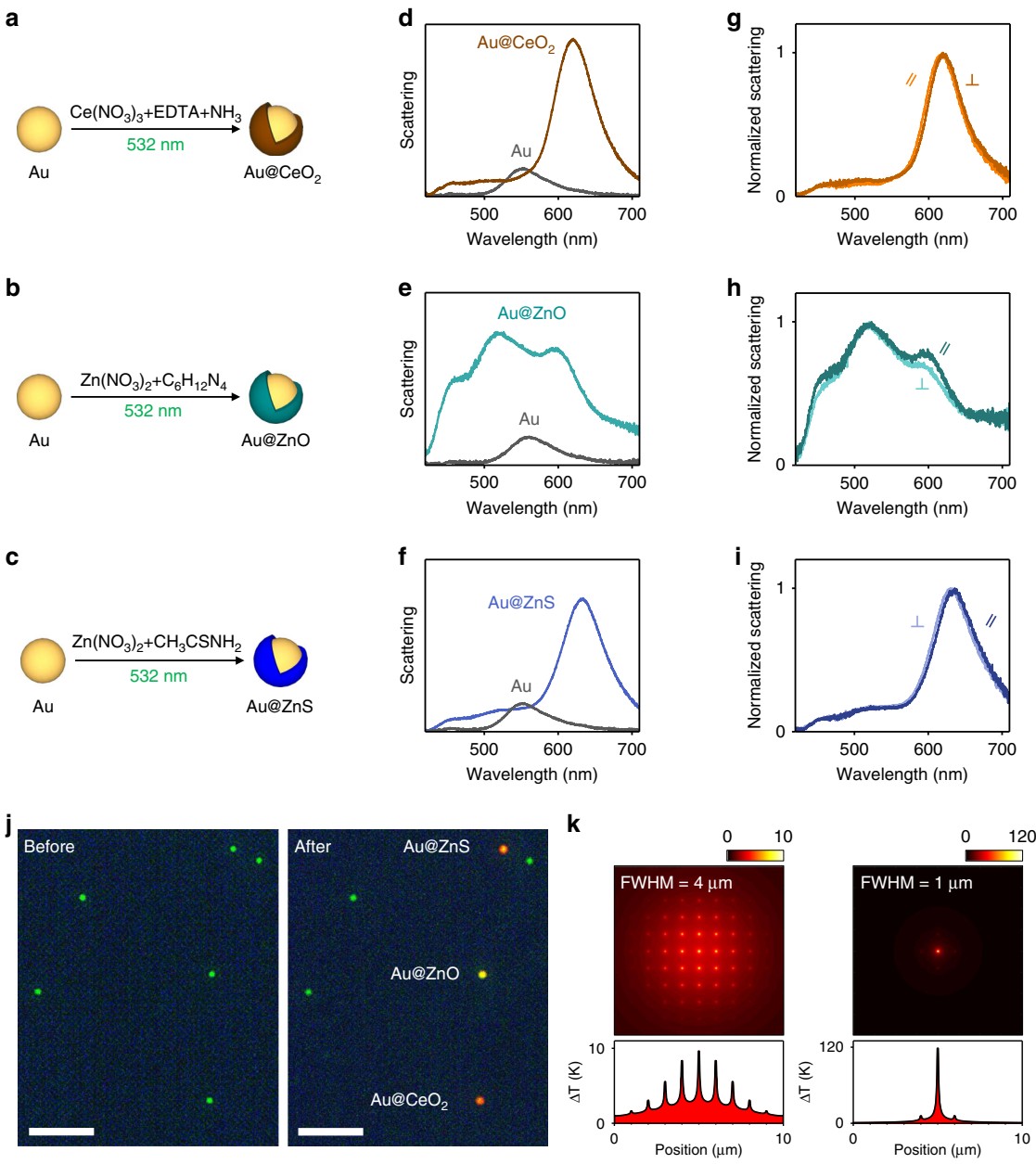

**Fig. 5 Photothermal growth of different semiconductors. a–c** Schematic representation of the photothermal synthesis of **a** Au@CeO₂, **b** Au@ZnO, and **c** Au@ZnS core@shell nanospheres. **d–f** Dark-field scattering spectra of single Au nanospheres before (gray) and after the photothermal growth of **d** CeO₂ (brown), **e** ZnO (teal), and **f** ZnS (blue) shells. **g–i** Scattering spectra of the Au@CeO₂, Au@ZnO, and Au@ZnS core@shell nanoparticles from **d–f** measured through a polarizing filter aligned both parallel and perpendicular to the 532 nm laser polarization. **j** Dark-field scattering images of Au nanoparticles deposited on a quartz substrate, before (left) and after (right) the photothermal growth of individual Au@ZnS, Au@ZnO, and Au@CeO₂ core@shell nanospheres. All green dots correspond to bare Au nanoparticles. Scale bars correspond to 10 μm. **k** Colormaps of the calculated distribution of the temperature increase **k** across an infinite square array of 66 nm Au nanoparticles with an interparticle spacing of 1 μm and illuminated with a 532 nm laser of 4 mW power possessing a beam spot with a full width at half maximum of 4 μm (left) and 1 μm (right). The bottom panels plot a crosscut along the centres of the maps.

90 °C using conventional heating generates micron-sized ZnO crystals after 1 h (Supplementary Note 16), while under similar conditions the ceria synthesis exhibits a slow shell growth of ~10 nm. In striking contrast to the needle-like ZnO morphologies grown in ensemble, photothermally grown ZnO shells are isotropic, similar to the case of CeO₂ (Fig. 5g), as revealed by polarization-dependent scattering measurements (Fig. 5h). These results suggest that light-induced nanoscale temperature gradients can circumvent spatial anisotropies driven by different facet reactivities[37], and can therefore be used to grow shape-controlled nanostructures, unattainable by conventional hydrothermal methods.

To further broaden the scope of our photothermal approach beyond the synthesis of metal oxides, we finally synthesize ZnS shells over individual Au nanospheres. The reaction is driven by the thermal decomposition of Zn-thioacetamide complexes to precipitate zinc sulfide[38] (see Methods). Upon photothermal ZnS shell growth during 15 min of laser irradiation, we observe an

LSPR red shift of 82 nm corresponding to a shell thickness of ~15 nm (Fig. 5f). Similar to the case of $CeO_2$ and ZnO, the ZnS shell growth is also isotropic (Fig. 5i).

Interestingly, both ZnO and ZnS exhibit intrinsic PL unlike $CeO_2$, thereby making it challenging to track the photothermal growth of these semiconductors. In the case of ZnO and ZnS shell growth, we observe a decreasing IE signal from the core@shell nanoparticle during the period of illumination (Supplementary Note 17). We hypothesize that the IE from the plasmonic nanoparticle is non-radiatively quenched by the transitions giving rise to the luminescence in these semiconductors.

Our localized-photothermal technique allows us to perform controlled growth of core@shell nanoparticles with different chemical compositions over a two-dimensional substrate. We demonstrate this capability by photothermally growing $CeO_2$, ZnO, and ZnS shells over three different individual Au nanoparticles deposited on the same substrate and separated by distances of a few microns. In this experiment, we focus the laser spot over three individual particles, while flowing the appropriate growth solutions one at a time. Between each growth step, the flow cell is flushed with ultrapure water to remove any remnant precursors. In Fig. 5j we show the dark-field scattering images of these nanoparticles before and after photothermal growth.

Given the spatial confinement of our synthetic method, one can envisage the use of nanoscale plasmonic reactors to pattern substrates with various complex hierarchical structures, such as core@shell@shell and bimetallic nanoparticles. For example, using our diffraction-limited technique on an array of metal nanoparticles, site-selective chemical reactions could be performed even for interparticle spacings of the order of 1 μm. In Fig. 5k, we show calculated temperature distributions for square arrays of Au nanoparticles with a diameter of 66 nm and an interparticle spacing of 1 μm illuminated using a 532-nm laser with an optical power of 4 mW. For a FWHM of the laser spot of 1 μm, the nanoparticle located at the center of the Gaussian beam heats up ~100 K more than its adjacent neighbors. Considering a chemical reaction with a typical activation energy of 50 kJ/mol[39], the nanoparticle at the centre of the beam can therefore photothermally drive chemical reactions with a rate that is two orders of magnitude faster compared to its nearest neighboring particles (see Supplementary Note 18).

In conclusion, we have shown how localized temperature gradients produced by plasmonic heating can be exploited to drive spatially confined chemical reactions. Such fine control over temperature spatial profiles are attained by focused illumination over individual nanoparticles, thereby preventing thermal homogenization due to collective heating effects. Even though we exploit photothermal effects to grow metal oxide and sulfide shells over individual Au nanoparticles, our method can be extended to the activation of any temperature-sensitive chemical reaction. In our work, we intentionally slow down the shell growth process by tuning both the chemical and optical parameters. This strategy allows us to follow the photothermal growth evolution in real time, while preventing any reaction in the dark. Photothermal temperature gradients, however, are typically activated over timescales of nanoseconds and faster growth kinetics can therefore easily be attained by tuning the laser irradiance, the concentration of the reactants, and the thermal conductivity of the environment. A faster shell growth would also allow to minimize thermal drifts during laser irradiation and therefore obtain a more straightforward relationship between the laser power and the final shell thickness than the one shown in Fig. 2d. Combining our technique with automated particle centering algorithms[40], one can envisage the development of fast printing of hierarchical plasmonic nanoparticles with new morphologies and advanced functionalities over large areas. Such

two-dimensional substrates with functional nanomaterials can have potential applications in the fields of photon upconversion[41], light-emitting diodes[42], and antenna-reactor catalytic complexes[43].

## Methods

**Synthesis of Au nanospheres**. Au nanospheres with a diameter of 66 nm are synthesized by a two-step seed mediated technique, using a previously reported method[44]. In brief, 5 mL of 0.5-mM $HAuCl_4$ and 5 mL of 200 mM CTAB aqueous solutions are mixed in a 25-mL round bottom flask, to which a 0.6 mL of 10-mM $NaBH_4$ is added in one-shot and stirred rapidly for 2 min. The solution is kept undisturbed at 27 °C for 3 h, to synthesize 10-nm Au nanoclusters. In another 100-mL round bottom flask, 40 mL of 100 mM cetyltrimethylammonium chloride (CTAC), 2.6 mL of 10 mM ascorbic acid and 0.1 mL of 10-nm Au nanoclusters are added in sequence and mixed well at 27 °C. To the above mixture, 40 mL of 0.5-mM $HAuCl_4$ is added using a syringe pump at a rate of 40 mL/h, under constant stirring. The final product is centrifuged thrice ($7200 \times g$, 10 min) and redispersed each time in 20-mM CTAB.

**Synthesis of Au nanorods**. $76 \times 25$-nm Au nanorods are synthesized by a two-step seed mediated technique, using a previously reported method[45]. In brief, 5 mL of 0.5-mM $HAuCl_4$ and 5 mL of 200-mM CTAB aqueous solutions are mixed in a 25-mL round bottom flask, to which a 1 mL of 6-mM $NaBH_4$ is added in one-shot and stirred rapidly for 2 min. The seed solution is aged at 27 °C for 30 min. To synthesize the nanorods, 1.8 g of CTAB and 0.7 g of KBr are mixed in a 100-mL round bottom flask by heating it to ~70 °C. The flask is then cooled to 30 °C for 15 min, Subsequently, 2.4 mL of 4-mM $AgNO_3$, 50 mL of 1-mM $HAuCl_4$ and 0.9 mL of 64-mM ascorbic acid are added in sequence, at intervals of 15 min. The above solution is vigorously stirred for 30 s. 0.16 mL of Au seeds is added to the above growth solution under stirring for 30 s, and then kept undisturbed overnight. The final product is centrifuged thrice ($16,000 \times g$, 10 min) and redispersed in 20-mM CTAB.

**Synthesis of Au@$CeO_2$ core@shell nanoparticles using conventional heating**. Au@$CeO_2$ core@shell nanoparticles are synthesized by adapting a previous report[22]. 5.45 mL of 25-mM CTAB and 50 μL of 66-nm Au nanospheres are mixed together in a 15 mL vial. 0.75 mL of EDTA-$NH_3$ solution and 0.075 mL of 100-mM $Ce(NO_3)_3$ solution is added to the above mixture, sequentially and mixed using a vortex-mixer. The above solution is then heated to 90 °C in an oven for 1 h. The EDTA-$NH_3$ solution is prepared by adding 0.1169 g of EDTA to 40 mL of water, followed by the addition of 0.38 mL of 30% $NH_4OH$ solution.

**Synthesis of Au@ZnO core@shell nanoparticles using conventional heating**. Au@ZnO core@shell nanoparticles are synthesized by adapting a previous report[37]. 1.2 mL of 25-mM CTAB and 50 μL of 66-nm Au nanospheres are mixed together in a 15 mL vial. 2.8 mL of deionized MilliQ water is added to the above while, followed by the addition of 0.3 mL of 0.1 M $Zn(NO_3)_2$ aqueous solution and 3 mL of 0.1 M HMT. The above mixture is vortex-mixed between each precursor addition. The above solution is then heated to 90 °C in an oven for 1 h.

**Synthesis of Au@ZnS core@shell nanoparticles using conventional heating**. Au@ZnS core@shell nanoparticles are synthesized by adapting a previous report[38]. 1.2 mL of 25-mM CTAB and 50 μL of 66-nm Au nanospheres are mixed together in a 15 mL vial. 2.8 mL of deionized MilliQ water is added to the above, followed by the addition of 0.3 mL of 0.1 M $Zn(NO_3)_2$ aqueous solution and 3 mL of 0.1 M thioacetamide (TAA). The above mixture is vortex-mixed between each precursor addition. The above solution is then heated to 90 °C in an oven for 1 h.

**Photothermal growth of core@shell nanostructures on a quartz substrate**. Photothermal growth of core@shell nanoparticles are performed inside a quartz flow cell. To fabricate the flow cell, a few microliters of a very diluted solution (optical density < 0.01) of Au nanoparticles is drop casted on to a quartz cover slip and allowed to dry. The coverslip is precleaned prior to nanoparticle drop casting, by treating it with 2 M KOH at 80 °C for 20 min, followed by multiple rinsing and sonication in water. Another quartz slide is drilled with 1 mm holes, for inserting plastic tubes (which act as the inlet and outlet for flowing the reaction mixture) into these holes and fixed with epoxy. The quartz slide and cover slip are then glued together with epoxy, to construct the flow cell. The liquid channel thickness is ~1 mm. The semiconductor growth solution is injected to the flow cell with a syringe pump at a rate of 20 μL/min during irradiation and spectroscopy. The solution used for the photothermal growth of $CeO_2$, ZnO, and ZnS shells around individual Au nanoparticles is identical to the corresponding solutions used in their ensemble synthesis in dark, except for the substitution of the suspension of Au nanoparticles with an equal volume of MilliQ water. Single Au nanoparticles are identified using their scattering spectra and irradiated with a focused 532-nm cw laser of varying optical power for 15 min to photothermally synthesize the core@shell nanoparticle.

**Photothermal growth of core@shell nanostructures on a TEM membrane**. To image the photothermally grown core@shell nanoparticles, a few microliters of a dilute suspension of Au NPs (optical density < 0.01) are dispersed on a plasma cleaned SiN membrane ($0.25 \times 0.25$ μm$^2$, 150-nm thick) bought from SPI supplies. The SiN membrane is glued on to the removable tip of a JEOL TEM sample holder using a two-part epoxy adhesive. A quartz slide drilled with 1 mm holes, for inserting plastic tubes, is attached to the TEM holder containing the SiN membrane using melted paraffin film, to assemble the flow cell. A syringe pump is used to flow the ceria growth solution at a rate of 20 μL/min. The in-plane position of gold nanoparticles close to the centre of the SiN membrane are recorded using the x,y coordinates of the corners of the membrane as our reference system. Selected nanoparticles at the centre are then irradiated using a focused 532 nm laser for 15 min. After irradiation, the flow cell is dismantled by gently pushing a scalpel between the quartz-paraffin film interface. Once the TEM sample holder containing the membrane is isolated, it is rinsed with MilliQ water, let to dry, and inserted in to the TEM. The coordinates of the SiN membrane corner are again used to identify the irradiated nanoparticles and their TEM images are taken.

**Instrumentation**. Single-particle dark-field imaging and scattering spectroscopy are performed on a Zeiss Axio Observer microscope equipped with a scanning x,y piezo-stage. Scattering spectra of nanoparticles are performed by illuminating them with an LED lamp focused through an EC epiplan 50× objective (N.A. = 0.75) via a dark-field reflector. The back scattered light from the nanoparticles are collected with the same objective and guided to the Andor Newton EMCCD coupled to the Shamrock 500i spectrometer. For 532-nm cw laser irradiation experiments, the laser is focused on to the gold nanoparticles using the same objective but using the bright-field reflector cube. The laser beam (CNI laser, MGL-FN-532) power is varied between 0 and 470 mW, to obtain various nanoparticle surface temperatures. To collect the IE, a 532 nm notch filter (Semrock, NF01-532U-25) is placed in the emission path, to avoid any laser damage of the spectrometer. The laser power at the sample is measured by removing the microscope objective lens, and directing the collimated light on to a Newport thermopile sensor connected to an optical power meter. The integration time of the PL spectra ranges between 5 and 30 s, depending on the laser illumination intensity. The transmission of the objective lens is corrected in the measured power at the sample, for estimating the nanoparticle surface temperatures. For ensemble optical measurements, we use a PerkinElmer Lambda 1050 UV−vis−NIR spectrophotometer. BF-STEM images and EDX maps are measured on an FEI Verios 460 scanning electron microscope equipped with a scanning transmission electron microscopy retractable detector.

**Mie calculations and FDTD simulations**. Mie calculations of spherical Au nanoparticles and Au@semiconductor core@shell nanoparticles in a homogeneous environment are performed using a MATLAB script[46]. The calculations are performed using the formalism given by Bohren and Huffman[10].

Lumerical FDTD simulations are performed to study the optical properties of Au and Au@semiconductor core@shell spheres/rods on a quartz substrate. In our simulations, the Au nanosphere/rod is placed on top of a quartz substrate, and the entire system is suspended in a background refractive index of 1.333 corresponding to water. For Au, we use the dielectric function measured by Johnson and Christy[47]. The refractive index of quartz is obtained from the supplier website, while a constant refractive index of 2.3 is used for the ceria shell. A total-field scattered-field source is used to illuminate the Au particles from beneath the quartz-nanoparticle interface at a distance of 600 nm from the nanoparticle. Absorption and scattering monitors kept at distances of 300 and 900 nm respectively, are used to record the nanoparticle cross-sections. The mesh region covers the entire nanoparticle and the mesh size of all simulations is kept at 0.5 nm. Perfectly matched layer (PML) absorbing boundaries are used to border the simulation region.

## Data availability

The data that support the findings of this study are available from the authors on reasonable request.

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

## Acknowledgements

This work was supported by the Dutch Research Council (Nederlandse Organisatie voor Wetenschappelijk Onderzoek) via the NWO Vidi award 680-47-550. We gratefully acknowledge Dr. Guillaume Baffou for his invaluable comments and suggestions. We are also grateful for insightful feedback received from Prof. Romain Quidant and from the members of the Nanomaterials for Energy Applications group at DIFFER.

## Author contributions

R.K. and A.B. conceived the idea for this project. R.K. performed all the experiments and the FDTD simulations. R.K. and G.K. performed the data analysis and wrote the first draft of the manuscript. All authors edited the manuscript. A.B. supervised the project.

## Competing interests

The authors declare no competing interests.
