## [Peer Review File · Nature Communications]

Reviewers' comments:

Reviewer #1 (Remarks to the Author):

Baldi and coworkers report on the synthesis of semiconductor shells around gold nanoparticle cores enabled exclusively by the heat generated locally at the individual particle level upon illumination with 532 nm laser light. The authors exclude the role of hot holes in this reaction and demonstrate that the proposed process affords the growth of uniform shells, which is not possible via regular heating of the reaction vessel. The study is interesting and extremely valuable at a time where the field is trying to unravel the interdependence of hot carriers and heat to carry out chemistry at the metal interface. However, in its current form it is still too preliminary for publication. I have explained my opinion in detail in the following:

1. What do the authors mean when on page 3 they write “individual”? Does it mean that their method does not really illuminate individual particles?
2. It is not clear to me if and how the temperature on the individual illuminated nanoparticles is measured. Also, it is not clear whether the neighboring particles are far enough on the substrate from the illuminated one to not experience heating due to heat transport from the illuminated particle. SEMs should be provided to show the actual particle to particle separation on the substrate. If particles are illuminated under a known sequence, is there a correlation between position and shell thickness/optical properties? Controls need to be set up to prove it.
3. To prove that hot holes are not at play here, the authors point to the band alignment and the mean free path length of holes in these structures. However, because these studies are at the single particle level, an estimation of band alignment for the exact crystalline structure of ceria produced should be reported, along with HR-TEM characterization, for this hypothesis to be verified.
4. All the thermal properties of the medium, substrate, etc. are calculated and not measured. While necessary, this calculation is however not sufficient, and local measurements need to be carried out.
5. Details on the FDTD calculations employed and described on page 8 need to be provided in the SI, so that is possible to evaluate the validity of the approach and verify how the role of the substrate has been taken into account, importantly because at least two different types of substrates have been employed in these studies.
6. A statistical assessment of optical and morphological properties of the nanoparticles at the population level (more than the 35 particles studied here) need to be reported to assess the validity of the approach and the analysis of the data.
7. In various plots the authors report the data points with colored spheres. These are however misleading as it is not clear what the magnitude of the error is in these measurements and therefore how accurate the fit is.
8. In equation 3 the authors propose a constant SERS response of the reactants on the nanoparticle surface. However, as the nanoparticle grows so does its surface area, which in turn affects the SERS signal. Furthermore, while the growing shell may impact the identity and intensity of the signal, this effect is in rapid evolution during the first growth stages. Even if we take into account only these two points, the equality assumed for the SERS term in Eq. 3 is not correct.

9. The experiments on the nanorods need to be integrated with laser irradiation tuned with the longitudinal LSPR as controls, with experiments under polarized light, and with an explanation on why growth is (or not) isotropic around the nanorod. I am also curious why the spectra in figure 4 are cut at 600 nm. TEM spectra for these additional nanostructures are needed.

In conclusion, while the study is interesting, it still needs to be completed with additional experiments before becoming publishable.

Reviewer #2 (Remarks to the Author):

In this paper the authors report an interesting approach to grow controlled composite nanoparticles using plasmonic photothermal effects. In particular, the authors demonstrate growth of CeO₂, ZnO and ZnS shells over Au nanoparticles and Au nanorods. The major novelty aspect consists in the possibility of controlling with great accuracy the growth of the oxide shell on each individual nanoparticle potentially achieving much higher uniformities than in colloidal processes where temperature cannot be well controlled at the nanoscale. The authors clearly present their methodology and report a substantial amount of convincing data. Overall, I think this work is very interesting and worth of publication in Nature Communication. Indeed, it is of interest for a broad community as it could enable the realization of highly controlled optical and chemically active substrates (e.g. for sensing, photochemistry, material science etc.). Nonetheless, I think the authors should address these following major concerns before the paper can be accepted for publication.

Major Points

- One aspect that I find a bit surprising is that the authors did several TEM images but do not appear to use TEM to measure the thickness of the semiconductor shell grown around the Au nanoparticles. Instead they rely purely on FDTD simulations. Although the FDTD simulations can be a good tool for covering the wide range of thicknesses explored by the authors, from Fig. 1d it is clear that the shell is quite rough and far from an idealized perfectly spherical shell. It would be thus very helpful if the authors could show the correspondence between the calculated shell and measured (TEM) shell thickness for a one or two shell thicknesses (ideally a thin one and a thick one encompassing the range of thicknesses studied). In particular, the TEM images would allow to assess the quality of the shell and the statistic distribution of its thickness. If this is not feasible it would be helpful if the authors could explain the experimental constraints that limit them to using solely FDTD.
- Related to the previous point, one of the major claims of the work is that this approach allows highly controlled growth of the semiconductor shell. In this respect, I think that the work would be greatly strengthened by showing some statistics on the reproducibility of the process. Indeed, it would be interesting to show that under XX mW/um² laser irradiation for YY min the shell growth is ZZ+/-zz nm. It could then be very helpful to relate the uncertainty to the experimental conditions: does it originate from the size dispersion of the initial nanoparticle size? Does it originate from the uncertainty in the laser/nanoparticle overlap? What conditions have the largest impact and should be thus controlled with the greatest precision? Such discussion would be very instructive for other researchers interested in applying this method.
- In the discussion of the plasmonic photothermal effect, the authors assume that convective heat

removal is negligible for their problem (i.e. heat is removed solely by diffusion through water which has a low thermal conductivity). However, this assumption is reasonable only if the liquid surrounding the nanoparticles is highly confined so that convective motion cannot reach large velocities. However, the authors do not discuss at all the geometry of their test cell (apart from showing a schematic without dimensions in Fig.1). It is thus very important that the authors provide greater details about their test cell. Also, in the main text they should explicitly address the assumption of negligible convection.

- Related to the previous point, the authors explain that the CeO₂ shell growth is a very good test process because it does not occur below 90C. This means that there should be a laser threshold power below which irradiation does not lead to any shell growth. This laser threshold would also enable the authors to validate their thermal model. Indeed, they should find that for such laser power and laser position the temperature increase calculated with eqn. 2 or eqn S9, which neglect convection and assume an average thermal conductivity for the surrounding medium, is below 90C. Do the authors observe such threshold? If not, is there any experimental limitation that hinders such observation? The authors might want to discuss this point in the manuscript.
- In Fig. 2e the authors show that the scattering signal is independent of polarization. However, they never discuss how sensitive is their scattering signal to the shell thickness. In fig. S6 they show the result of numerical calculations but what about the experimental result, which has a broader plasmon linewidth and a noisier signal? If the grown shell was differing by only a few nm, would they be able to distinguish it in the DF signal? What about the polarization dependent measurements for the Au nanorods? Connecting to my first point, TEM images might clarify this aspect. In particular, Au nanorods would be a better test because the asymmetry should be maximum if other effects (e.g. near field, hot carriers) were to play any role.

Minor Points

- I feel that the data repartition between the main text and the supplementary often does not reflect the really important points that the authors are trying to make in the manuscript.
 - o For example, in Fig. 1 most of the space is taken by the optical set-up description and only a minor part is devoted to the data/TEM/EDX. Although in this work the simultaneous measurement of diverse optical signals is important, I find that the optical set-ups are not novel and could be very well described in detail in supplementary without loss of understanding from the reader. Instead, Fig. 1 has no clear sketch that explain the data shown in Fig1c-d and more broadly what the authors are doing. It is true that in the experimental set-up drawings (Fig1a-b) the authors have added a shell around an Au nanoparticle under laser irradiation but the detail is practically invisible for the reader. In fact, the attention is entirely focused on the large optical set-up schematic. I thus suggest that they zoom in onto the sample schematic to clearly show the nanoparticle without/with shell without/with irradiation, respectively. They can simply use some labels (e.g. Dark Field, PL) under such zoomed view to explain which optical signals they monitor, moving the detailed optical path to SI. This schematic would then make clear the data that are currently present in Fig.1 as well as the overall concept presented in this work.
 - o Another example is Fig. 3. Most of the space is devoted to the analysis of the PL signal while very little space is given to the final result (panel d). Although I totally understand that the authors are using a novel approach to analyze the signal and thus want to clarify their process to the reader, I wonder

whether the technical details might take less space in the manuscript figure in favor of fewer, clearer experimental results. In this respect, the long and complex time traces of PL could go to the SI while only a few time profiles (currently shown as tiny sketches in Fig. 3) could be shown in the manuscript with much greater clarity for the reader.

Overall, these are just suggestions. Yet, I invite the authors to think carefully how their figures could be improved/restructured to avoid diluting their major experimental results among the description of smaller technical details that could be instead reported in the supplementary information.

Reviewer #3 (Remarks to the Author):

The reviewed manuscript presents a comprehensive experimental analysis of light-driven nanoparticle growth. Specifically, the authors investigate the growth of dielectric and semiconductor shells on metallic nanoparticles induced by the localized plasmon heating. Using dark field microscopy, the authors are able to address and characterize this phenomenon at the single particle level, being able, at the same time, to temporally follow the shell growth. Thanks to that, the authors present very clean and clear results that allow them to draw solid conclusions. In my opinion, the paper is very interesting for the broad community of nanophotonics, timely, and well-written, and thus merits publication. That said, I do have a few comments and suggestions that, once addressed, I believe can help to improve the manuscript:

- At the very end of page 3, the authors mention “inelastic scattering.” Although later in the paper the meaning of this becomes clear, I think the authors should expand a bit more on what they mean by that on page 3.

- On page 8 and Supplementary S9, the authors explain that they obtain an effective thermal conductivity by taking the average of the water and quartz thermal conductivities. However, it is not clear to me that this is justified. I believe that most of the surface of the nanoparticle should be covered by water, furthermore, the part that is in contact with the substrate should actually be experiencing a contact thermal resistance that may be very different from that predicted by the bulk thermal conductance, as shown in recent papers [1]. I recommend the authors to elaborate more on how the accuracy on the estimation of the thermal conductance can impact their results.

- On page 11, the authors claim that the change on photoluminescence can be assumed to be proportional to the change on scattering cross-section. However, does the temperature change impact these two quantities in the same way?

- The authors show that the growth of the different shells is independent of the polarization of light. This is consistent with the fact that the temperature in the nanoparticle is rather uniform due to the high thermal conductance of metals. Although an isotropic growth is probably very useful in many cases, sometimes a non-homogeneous growth may be desired. Could the authors envision any potential

approach leading to a non-homogenous growth? Perhaps using ultrafast illumination?

[1] P. Zolotavin, A. Alabastri, P. Nordlander and D. Natelson, Plasmonic Heating in Au Nanowires at Low Temperatures: The Role of Thermal Boundary Resistance, *ACS Nano*, 10(7), pp. 6972-6979, (2016).

Reviewer #1 (Remarks to the Author):

Baldi and coworkers report on the synthesis of semiconductor shells around gold nanoparticle cores enabled exclusively by the heat generated locally at the individual particle level upon illumination with 532 nm laser light. The authors exclude the role of hot holes in this reaction and demonstrate that the proposed process affords the growth of uniform shells, which is not possible via regular heating of the reaction vessel. The study is interesting and extremely valuable at a time where the field is trying to unravel the interdependence of hot carriers and heat to carry out chemistry at the metal interface. However, in its current form it is still too preliminary for publication. I have explained my opinion in detail in the following:

We thank the reviewer for appreciating the significance of our study.

1. What do the authors mean when on page 3 they write “individual”? Does it mean that their method does not really illuminate individual particles?

This confusion stems from our intention to use quotes to stress the importance of illuminating individual nanoparticles when using nanoscale temperature gradients for nanomaterial synthesis. We have removed the quotes to avoid any misunderstanding: we do indeed illuminate one particle at a time. We have also changed the sentence

In a typical experiment, we focus a Gaussian laser beam of $\sim 2 \mu\text{m}$ in diameter to excite the gold nanoparticles.

into

In a typical experiment, we focus a Gaussian laser beam of $\sim 2 \mu\text{m}$ in diameter to excite a single gold nanoparticle.

2. It is not clear to me if and how the temperature on the individual illuminated nanoparticles is measured.

In our experiments, we do not directly measure the temperature but we estimate it using equation 2: please see the first paragraph of the section “Tunable and isotropic shell growth” for a detailed explanation of the procedure.

While it would be ideal to know the exact temperature of our nanoparticles, measuring the temperature of a single plasmonic nanoparticle under laser illumination in a liquid environment and during a shell growth reaction is very challenging. Typically temperature measurements on illuminated single plasmonic nanoparticles have been performed using temperature-sensitive chemical probes and measuring their fluorescence or Raman signals, which can be calibrated to samples heated in the dark¹. Employing such chemical probes in our experiments may however affect the photochemical growth of our core@shell nanostructures and can also interfere with the *in situ* photoluminescence measurements employed to study their growth kinetics. Distinguishing the

weak photoluminescence signal of nanoparticles from the optical signal of these chemical probes would also be complex, if at all possible.

Another strategy that has been suggested to measure the nanoparticle surface temperature is by analyzing the anti-Stokes signal of the photoluminescence of the plasmonic nanoparticles². However, recently it has been found that such analysis reveals the electron temperature, which matches the nanoparticle temperature only at mild laser intensities, far lower than in our experimental conditions³.

A promising route would be to use thermal imaging quadriwave lateral shearing interferometry (TIQSI), in which the change in refractive index of the surrounding water medium is used to estimate the surface temperature of the illuminated nanoparticles⁴. This technique typically requires a static system and it is therefore challenging to implement in our reactive conditions. Nevertheless, in a new collaborative research project with dr. Guillaume Baffou (CNRS, Marseille, France), we are exploring the possibility of measuring the temperature of these nanoparticles using TIQSI. The first results of this collaboration look promising and we find that under a laser intensity of $2.9 \text{ mW}/\mu\text{m}^2$, we measured a surface temperature of 170 K, which is consistent with our estimated maximum temperature of 199 K.

We would like to stress out, however, that the main message of the paper, as we write in our abstract, is to “demonstrate how localized plasmonic photothermal effects can generate spatially confined nanoreactors by activating, controlling, and spectroscopically following the growth of individual metal@semiconductor core@shell nanoparticles”. For this purpose, one does not necessarily need to know the exact nanoparticle temperature and our estimated values are sufficient to demonstrate our proof of principle.

Also, it is not clear whether the neighboring particles are far enough on the substrate from the illuminated one to not experience heating due to heat transport from the illuminated particle. SEMs should be provided to show the actual particle to particle separation on the substrate.

In a typical illumination experiment, we drop cast a very dilute suspension of Au nanoparticles on a quartz substrate in order to ensure that the average interparticle spacing is larger than $5 \mu\text{m}$, which is evident from the representative optical images in Fig 5j (scale bars = $10 \mu\text{m}$) and in Supplementary Fig 5 (in which we have now added the scale bar).

We illuminate our particles with a Gaussian laser beam of spot size $\sim 2 \mu\text{m}$. Assuming an interparticle distance of $\sim 5 \mu\text{m}$, the temperature increase on the nanoparticle adjacent to the one being illuminated is ~ 150 times lower⁵. For a temperature increase of 200 K on the illuminated nanoparticle, the adjacent nanoparticle will then experience a negligible temperature increase of ~ 1 K, far too small to drive the growth of any semiconducting shell. This is clearly shown, for example, by Figure 5j in which the scattering properties of the Au particle adjacent to the one irradiated during the growth of a ZnS shell (top right corner of the images) remain unchanged.

Optical images are sufficient to measure the interparticle distance since we deposit near monodisperse gold particles of 66 nm in diameter (Supplementary Fig. 1) that are very bright light scatterers in the visible and are therefore easily detected in our dark-field microscope setup. Moreover, our light scattering experiments require transparent and flat substrates that are insulating and therefore unsuited for electron microscopy.

If particles are illuminated under a known sequence, is there a correlation between position and shell thickness/optical properties? Controls need to be set up to prove it.

There is no particular sequence, but each new experiment is done on a single nanoparticle that is at least 5 μm apart from the previous one. This ensures that no correlation exist between the particle position and the shell thickness. The control experiment is displayed in Supplementary Fig. 7.2 in which we plot the scattering spectra of nanoparticles adjacent ($> 5 \mu\text{m}$) to a particle that has been illuminated with the 532 nm cw laser, before and after laser irradiation. Note that the resolution of our spectrometer allows us to determine the plasmon peak position with a typical accuracy of ± 0.01 eV, which corresponds to an uncertainty on the ceria shell thicknesses of less than 1 nm (see also our reply to comment #5 of reviewer #2). The absence of any plasmon shift in Supplementary Fig. 7.2 is a clear evidence that we do not generate any temperature increase or shell growth on particles adjacent to the illuminated one.

To clarify this point, in the description of supplementary figure 7.2 we have modified the following text:

The LSPR of Au nanoparticles remain unchanged after flowing the ceria growth solution for 30 min in the absence of laser irradiation, confirming no changes to the nanoparticle surface. Since these particles are kept in the dark, we do not generate any photothermal heating, which prevents ceria shell growth.

into:

In Figure 7.2 we plot the scattering spectra of Au nanoparticles adjacent ($> 5 \mu\text{m}$) to a nanoparticle that is irradiated, before and after laser irradiation. The LSPR of the Au nanoparticles that are not irradiated remains unchanged after flowing the ceria growth solution for 30 min, confirming no changes to the nanoparticle surface and no ceria shell growth.

3. To prove that hot holes are not at play here, the authors point to the band alignment and the mean free path length of holes in these structures. However, because these studies are at the single particle level, an estimation of band alignment for the exact crystalline structure of ceria produced should be reported, along with HR-TEM characterization, for this hypothesis to be verified.

The reviewer is of course correct that for an exact knowledge of the band alignment one would have to know the exact crystalline structure of the ceria. However, it is well known that colloidal CeO_2 syntheses lead to amorphous or nanocrystalline structures because of the low temperature at which the reactions are conducted and that crystalline ceria is only obtained upon annealing at temperatures above $\sim 500^\circ\text{C}$ ⁶. The band alignment we use in our reasoning has been experimentally measured on amorphous cerium dioxide films in a $\text{Au/CeO}_2/\text{Au}$ system⁷ and utilized to interpret photo-catalytic reactivity of Au@CeO_2 nanoparticles synthesized using a conventional hydrothermal approach similar to the one we use here⁸. We therefore feel confident in our suggested

interpretation. We have modified the initial strong wording of our Supplementary section S8 to better reflect the basis of our reasoning. In particular, we have modified the following text:

Supplementary S8 | Eliminating hot-hole contribution to Au@CeO₂ core@shell growth.

We rule out the contribution of hot-holes generated under plasmon excitation in driving the Au@CeO₂ core@shell nanoparticles, by correlating the Fermi level of Au and the band diagram of CeO₂¹.

into

Supplementary S8 | Hot-hole contribution to Au@CeO₂ core@shell growth.

The potential contribution of hot-holes generated under plasmon excitation in driving the Au@CeO₂ core@shell nanoparticles is analyzed, by correlating the Fermi level of Au and the band diagram of CeO₂¹.

We have also modified the last line of the Supplementary S8:

These considerations safely allow us to rule the contributions of hot holes in the plasmon-driven Au@CeO₂ core@shell nanoparticle synthesis.

into

These considerations safely allow us to rule out any significant contribution of hot holes to the plasmon-driven synthesis of Au@CeO₂ core@shell nanoparticles.

It should be noted that we perform our laser irradiation experiments on nanoparticles dispersed onto quartz coverslips that are then assembled into a flow cell using epoxy glue. Acquiring high resolution TEM images or electron diffraction patterns on individual nanoparticles on such insulating substrates is impossible (please also see our reply to comment #1 of Reviewer #2).

Finally, the synthesis of ZnO and ZnS shells does not involve any redox reaction, further confirming our proposed interpretation of the dominant role of photothermal effects.

4. All the thermal properties of the medium, substrate, etc. are calculated and not measured. While necessary, this calculation is however not sufficient, and local measurements need to be carried out.

We are puzzled by this comment, as measuring the thermal properties locally (i.e. at the nanoscale, in a liquid flow cell, under reactive chemical conditions) is both experimentally challenging, if not impossible, and not needed in the context of our work. All previous reports in the literature on localized heating of nanostructures utilize bulk thermal properties of the medium and the substrate

and have demonstrated that these values are sufficiently accurate to calculate photothermal effects at the nanoscale⁹⁻¹⁵.

5. Details on the FDTD calculations employed and described on page 8 need to be provided in the SI, so that is possible to evaluate the validity of the approach and verify how the role of the substrate has been taken into account, importantly because at least two different types of substrates have been employed in these studies.

All the reported spectra are collected from nanoparticles deposited on a quartz substrate. The SiN membrane is only used for electron microscopy. All necessary details of the FDTD simulations are already reported in the Methods section. We have made this clearer in the manuscript by changing the following line:

From the measured LSPR of the Au nanoparticle, we extract its radius R , and its absorption cross-section σ_{abs} , using finite-difference time-domain (FDTD) calculations and assuming a perfect spherical shape of the nanoparticle supported on quartz.

to

From the measured LSPR of the Au nanoparticle, we extract its radius R , and its absorption cross-section σ_{abs} , using finite-difference time-domain (FDTD) calculations (see Methods) and assuming a perfect spherical shape of the nanoparticle supported on quartz.

6. A statistical assessment of optical and morphological properties of the nanoparticles at the population level (more than the 35 particles studied here) need to be reported to assess the validity of the approach and the analysis of the data.

The aim of our work is to demonstrate that plasmonic photothermal effects can be used to grow different semiconducting nanostructures in a controlled way. In this respect, 35 single particle experiments in our opinion are more than sufficient to demonstrate our point. In the words of reviewer #2, we believe we already “report a substantial amount of convincing data”.

While we agree that more is better, each of these experiments is highly time consuming. For each experiment we select the nanoparticle to irradiate, measure its plasmon resonance with and without polarizers, switch the optical geometry of our microscope from dark-field spectroscopy to photoluminescence, irradiate the particle for 15 minutes, switch back to dark-field spectroscopy and measure the plasmon resonance of the final core@shell nanoparticle. Each nanoparticle experiment takes at least 1 hour of operations, without considering sample and solutions preparation, flow cell assembly, optics alignment, thermal equilibration of the setup, etc. Furthermore, 30% of the experiments typically fail and are discarded due to thermal instabilities in the laboratory leading to too large drifts of our samples during laser irradiation.

We would like to stress that nowhere in our paper we fit our measured plasmon resonance shifts or estimated photothermal temperature increases using a model to extract physical quantities. As such,

having a statistically significant number of single-particle experiments “at the population level” (supposedly of the order of hundreds or thousands), while being unfeasible experimentally, is also beyond the scope of our work.

7. In various plots the authors report the data points with colored spheres. These are however misleading as it is not clear what the magnitude of the error is in these measurements and therefore how accurate the fit is.

We are a bit puzzled by this comment: nowhere in our paper we fit the data to any particular model. The plots to which the reviewer is referring to are:

- Figure 2d: our measured plasmon resonances have typical full widths at half maximum smaller than 0.3 eV and the peak position is determined with a typical spectral accuracy of ± 0.01 eV. The error on the peak shift in the y-axis is therefore well within the size of the colored spheres. The temperature on the x-axis is estimated analytically as we mentioned already in the answer to comment #2. The error on the temperature values is challenging to estimate accurately, due to uncertainties in the thermal drift of the nanoparticles with respect to the laser spot and the error in the derivation of the absorption cross-section of the nanoparticle, which is modeled as a perfect sphere.
- Figures 2f and 3c: the red-shifts are measured in the same way as for figure 2d. The error bars are well within the size of the colored spheres. As we write in the figure captions, however, the black line are not fits to the data, but 1:1 correlations.

8. In equation 3 the authors propose a constant SERS response of the reactants on the nanoparticle surface. However, as the nanoparticle grows so does its surface area, which in turn affects the SERS signal. Furthermore, while the growing shell may impact the identity and intensity of the signal, this effect is in rapid evolution during the first growth stages. Even if we take into account only these two points, the equality assumed for the SERS term in Eq. 3 is not correct.

We agree with the reviewer. We have included the SERS term in equation (3) only for completeness as, in general, the inelastic emission can contain SERS contributions. In our experimental inelastic scattering spectra of Fig.3a, however, there are no noteworthy Raman signals of the reactants or of the products, except for the water signal at ~ 630 nm (which is cut off from the spectra). This is likely due to the weak near-fields at the Au nanoparticle surface compared to better SERS substrate materials like Ag. As such, the SERS term does not really affect our analysis and can be simplified in equation (3). Of course, if SERS signals are present, one has to be careful on how the LSPR peak is extracted out of the PL signal. Even in this case, however, we predict that the LSPR signal would be easy to extract since Raman signals typically possess much narrower peaks.

We have made this clearer in the manuscript by changing the following line:

The measured inelastic scattering spectrum is a superposition of different contributions, including the PL from the growing core@shell nanoparticle, the luminescence of the underlying quartz substrate, and the surface-enhanced Raman scattering of reactants on the nanoparticle surface. To

isolate the nanoparticle PL from the other contributions, we subtract all the IE spectra recorded over the course of 15 minutes of illumination with the first IE spectrum measured at time $t = 0$, according to (see also discussion in Supplementary S11):

to

The measured inelastic scattering spectrum is a superposition of different contributions, including the PL from the growing core@shell nanoparticle, the luminescence of the underlying quartz substrate, and the surface-enhanced Raman scattering (SERS) of reactants on the nanoparticle surface. In our experiments we don't see evidence of any SERS contribution, most likely due to the weak near-field enhancement of our gold nanoparticles. Assuming negligible SERS signal, we isolate the nanoparticle PL from the other contributions by subtracting all the IE spectra recorded over the course of 15 minutes of illumination with the first measured IE spectrum, according to (see also discussion in Supplementary S11):

Also, in Supplementary S11, we replace the following text

Using this approach, we remove background contributions from the underlying quartz substrate and the Raman signal of reactants, and isolate the photoluminescence of the plasmonic nanoparticle.

with

Using this approach, we remove background contributions from the underlying quartz substrate and the (negligible) Raman signal of reactants, and isolate the photoluminescence of the plasmonic nanoparticle.

9. The experiments on the nanorods need to be integrated with laser irradiation tuned with the longitudinal LSPR as controls, with experiments under polarized light, and with an explanation on why growth is (or not) isotropic around the nanorod. I am also curious why the spectra in figure 4 are cut at 600 nm. TEM spectra for these additional nanostructures are needed.

We are not sure about the additional knowledge these proposed experiments would provide. Our 532 nm laser is already polarized and our intention with the nanorod experiment is simply to show the flexibility of the photothermal growth: regardless of the shape of the plasmonic nanoparticle, photothermal effects can drive thermally-driven reactions at the nanoscale.

We cut the spectra at 600 nm for centering the plasmon resonance and for providing better aesthetics to the figure: there is no additional data in the cut-off parts of the spectra as can be seen in the plots below that reproduce Figure 4b and 4c with an extended wavelength range (the orange rectangle denotes the spectral range shown in the main text):

We have added the full spectral range of the dark-field scattering spectra of Au and photothermally grown Au@CeO₂ nanorods in Supplementary S13.

Note that, as can be seen in the scattering spectra above, the width of the chosen nanorods (~25 nm) is too small to lead to detectable scattering of the transverse mode. For this reason, we cannot measure if the ceria shell growth is isotropic (see also our reply to comment #5 of reviewer #2). We have added the underlined text to the manuscript:

In Fig. 4b, we plot the longitudinal LSPR of the Au NR measured before and after irradiation. Due to the narrow width of the nanorod (~25 nm), the transverse plasmon resonance, which is expected at ~500 nm, is not visible in our measured scattering spectra (see Supplementary S13).

Finally, as mentioned earlier, performing TEM images on these samples is not possible due to the presence of an insulating substrate assembled into a liquid flow cell and not necessarily needed for the main message of our work. As we argue in our reply to comment #5 of reviewer #2, the scattering spectra of these nanorods are a powerful tool to study the changes occurring on the nanoparticle surface.

In conclusion, while the study is interesting, it still needs to be completed with additional experiments before becoming publishable.

We believe the additional evidence provided and the reasonings above will be able to convince the reviewer that our paper is suitable for publication.

Reviewer #2 (Remarks to the Author):

In this paper the authors report an interesting approach to grow controlled composite nanoparticles using plasmonic photothermal effects. In particular, the authors demonstrate growth of CeO₂, ZnO and ZnS shells over Au nanoparticles and Au nanorods. The major novelty aspect consists in the possibility of controlling with great accuracy the growth of the oxide shell on each individual nanoparticle potentially achieving much higher uniformities than in colloidal processes where temperature cannot be well controlled at the nanoscale. The authors clearly present their methodology and report a substantial amount of convincing data. Overall, I think this work is very interesting and worth of publication in Nature Communication. Indeed, it is of interest for a broad community as it could enable the realization of highly controlled optical and chemically active substrates (e.g. for sensing, photochemistry, material science etc.).

Thank you for your kind words.

Nonetheless, I think the authors should address these following major concerns before the paper can be accepted for publication.

Major Points

1) One aspect that I find a bit surprising is that the authors did several TEM images but do not appear to use TEM to measure the thickness of the semiconductor shell grown around the Au nanoparticles. Instead they rely purely on FDTD simulations. Although the FDTD simulations can be a good tool for covering the wide range of thicknesses explored by the authors, from Fig. 1d it is clear that the shell is quite rough and far from an idealized perfectly spherical shell. It would be thus very helpful if the authors could show the correspondence between the calculated shell and measured (TEM) shell thickness for a one or two shell thicknesses (ideally a thin one and a thick one encompassing the range of thicknesses studied). In particular, the TEM images would allow to assess the quality of the shell and the statistic distribution of its thickness. If this is not feasible it would be helpful if the authors could explain the experimental constraints that limit them to using solely FDTD.

Correlated optical and electron microscopy measurements is certainly a very powerful technique, but it is quite challenging to implement especially in our experimental geometry (see also our reply to the third comment of reviewer #1).

We illuminate individual Au nanoparticles deposited on a quartz coverslip, which is assembled into a flow cell using epoxy glue. The use of epoxy prevents any leakage while flowing the semiconductor growth solution, but also prevents us from disassembling the flow cell at the end of the experiment in a clean, non-destructive, and reproducible way. Even if one were able to recover the substrate containing the core@shell nanoparticles, electron microscopy characterization on a quartz substrate requires the addition of a conducting layer (spin-coated conducting polymer or deposited thin metal film) resulting in images with rather low spatial resolution, typically of the order of 10 nm or worse.

This is the reason why we used an ultrathin (and fragile) silicon nitride membrane to visualize the Au@CeO₂ nanoparticles in Fig. 1. However, these substrates are very difficult to assemble and disassemble into a flow cell without breakage (we had to perform 6 experiments to get one intact

membrane at the end of the procedure). Issues such as the occurrence of air bubbles while flowing the precursor solution are common and cannot be removed from the membrane unless a high flow is applied, often leading to membrane breaking.

On the contrary, measuring the plasmon resonance shift of a Au@CeO₂ nanoparticle is relatively straightforward, non-invasive, and leads to a shell thickness accuracy better than 1 nm, if the refractive index of the surrounding semiconducting shell is known.

We incorporate this message in the manuscript where we insert the following underlined text:

No CeO₂ shell growth is observed for nanoparticles that are on the same TEM membrane but are not irradiated, in agreement with our optical scattering studies (Supplementary S7).

In principle, correlated optical-electron microscopy could be used to determine the final shell thickness of our plasmonically grown Au@CeO₂ nanoparticles. However, the use of ultrathin, fragile TEM membranes in liquid flow cells is experimentally challenging and cannot be relied upon to obtain significant statistics. We therefore base our shell thickness derivation upon the measured spectral shifts in the irradiated nanoparticles using dark-field scattering spectroscopy.

As an additional control experiment, we also measured the dark-field spectra of Au nanoparticles before and after laser irradiation in the absence of Ce³⁺ ions. The spectra perfectly overlap, indicating that the nanoparticles are stable under our laser irradiation intensities (Supplementary S7).

2) Related to the previous point, one of the major claims of the work is that this approach allows highly controlled growth of the semiconductor shell. In this respect, I think that the work would be greatly strengthened by showing some statistics on the reproducibility of the process. Indeed, it would be interesting to show that under XX mW/um² laser irradiation for YY min the shell growth is ZZ+/-zz nm. It could then be very helpful to relate the uncertainty to the experimental conditions: does it originate from the size dispersion of the initial nanoparticle size? Does it originate from the uncertainty in the laser/nanoparticle overlap? What conditions have the largest impact and should be thus controlled with the greatest precision? Such discussion would be very instructive for other researchers interested in applying this method.

This is an interesting point. We believe that the reproducibility of the method is already confirmed by Fig. 2d in which we have included the results from single particle illumination results taken from 5 sets of experiments performed on different days, in different flow cells, and using newly prepared reaction mixtures (as indicated with different colors in the figure below).

We have added the underlined sentence in the caption of Figure 2d:

(d) Red shift of the scattering spectra of 35 different single Au@CeO₂ core@shell nanoparticles after the laser-driven shell growth, plotted as a function of the calculated initial nanoparticle temperature increase. To ensure reproducibility, the data points have been collected over five different sets of experiments using newly prepared flow cell assemblies and reactant solutions. The dashed line at 210 K represents the bubble formation temperature typically observed under nanoparticle photothermal heating²⁸.

The scatter in the data is mostly due to the uncertainty in the temperature estimation, which in turn is due to the error in determining the nanoparticle position in the x-y plane with respect to the center of the Gaussian laser beam. This error is mostly due to slow thermal drift of the microscope sample stage during the 15 minutes of laser irradiation. We specify this point when we write:

The scatter in the data in Fig. 2d mainly originates from the error in the estimated temperature, due to the small drift in the nanoparticle position during the 15 minutes of laser irradiation.

Unfortunately, the scatter in the data prevents us from deriving a univocal relationship between the illumination power and the shell thickness in our experimental conditions. We would like to stress that in our experiments, we have intentionally slowed down the reaction rate by adding a large amount of the Ce³⁺ complexing agent EDTA, to be able to follow the reaction in real time and properly differentiate between photothermal driven processes and inherent thermal reactivity at ambient temperature in the dark. Such slow reaction rate makes us particularly sensitive to any drift during the 15 minutes of laser illumination. In practical applications, we can easily envision a chemical environment that allows a shell growth reaction to be completed in a much shorter time. We discuss this possibility in our conclusions, where we have now added the underlined sentence:

In our work, we intentionally slow down the shell growth process by tuning both the chemical and optical parameters. This strategy allows us to follow the photothermal growth evolution in real-time, while preventing any reaction in the dark. Photothermal temperature gradients, however, are

typically activated over timescales of nanoseconds and faster growth kinetics can therefore easily be attained by tuning the laser irradiance, the concentration of the reactants, and the thermal conductivity of the environment. A faster shell growth would also allow to minimize thermal drifts during laser irradiation and therefore obtain a more straightforward relationship between the laser power and the final shell thickness than the one shown in Figure 2d. Combining our technique with automated particle centering algorithms⁴⁰, one can envisage the development of fast printing of hierarchical plasmonic nanoparticles with new morphologies and advanced functionalities over large areas. Such two-dimensional substrates with functional nanomaterials can have potential applications in the fields of photon upconversion⁴¹, light-emitting diodes⁴², and antenna-reactor catalytic complexes⁴³.

The uncertainty in the initial nanoparticle size is a smaller contribution to the scatter of the data in Figure 2d, since we start with a monodisperse sample (Supplementary S2, standard deviation of 5%) and most of the nanoparticles that are selected for illumination have an initial plasmon resonance close to 555 nm.

3) In the discussion of the plasmonic photothermal effect, the authors assume that convective heat removal is negligible for their problem (i.e. heat is removed solely by diffusion through water which has a low thermal conductivity). However, this assumption is reasonable only if the liquid surrounding the nanoparticles is highly confined so that convective motion cannot reach large velocities. However, the authors do not discuss at all the geometry of their test cell (apart from showing a schematic without dimensions in Fig.1). It is thus very important that the authors provide greater details about their test cell. Also, in the main text they should explicitly address the assumption of negligible convection.

This is an interesting point which we did not properly address in our first version. Our flow cells are assembled using quartz slides glued to cover slips with epoxy and the flow cell thickness is therefore macroscopic (~1 mm) with respect to the dimensions of the nanoparticles. We have added the underlined sentence in the Methods section:

Photothermal growth of core@shell nanoparticles are performed inside a quartz flow cell. To fabricate the flow cell, a few microliters of a very diluted solution (optical density < 0.01) of Au nanoparticles is drop casted on to a quartz cover slip and allowed to dry. The coverslip is precleaned prior to nanoparticle drop casting, by treating it with 2M KOH at 80 °C for 20 min, followed by multiple rinsing and sonication in water. Another quartz slide is drilled with 1 mm holes, for inserting plastic tubes (which act as the inlet and outlet for flowing the reaction mixture) into these holes and fixed with epoxy. The quartz slide and cover slip are then glued together with epoxy, to construct the flow cell. The liquid channel thickness is ~1 mm.

Despite the macroscopic size of the liquid channel, the effect of convection on the temperature distribution around the nanoparticle is however negligible, as stated in this passage from the book “Thermoplasmonics: Heating Metal Nanoparticles Using Light”¹⁶:

“In liquids, heating gold nanoparticles may produce a thermal-induced fluid convection, as represented below (see the upward fluid velocity lines). However, due to the weak Rayleigh number of water, thermal-induced fluid convection is not supposed to distort the temperature distribution within the liquid, regardless of the temperature increase or the size of the heat source. Consequently, numerical simulations of temperature distribution in fluids can be performed without any consideration on thermal-induced fluid convection.”

We have added the following underlined sentence to the main text of our paper:

The nanoscale radius of curvature of the Au nanoparticles, in fact, allows superheating of water above its standard boiling point, as large Laplace pressures are required to sustain nanoscopic bubble formation²⁸. Our nanoscale plasmonic reactors therefore effectively act as optically-driven analogues to hydrothermal bombs, which are often used for the synthesis of high quality, single-crystalline materials. Note that the temperature distribution around the irradiated plasmonic nanoparticle is not distorted by the presence of photothermal convective flows, thanks to the low Rayleigh number of water²⁹.

4) Related to the previous point, the authors explain that the CeO₂ shell growth is a very good test process because it does not occur below 90C. This means that there should be a laser threshold power below which irradiation does not lead to any shell growth. This laser threshold would also enable the authors to validate their thermal model. Indeed, they should find that for such laser power and laser position the temperature increase calculated with eqn. 2 or eqn S9, which neglect convection and assume an average thermal conductivity for the surrounding medium, is below 90C. Do the authors observe such threshold? If not, is there any experimental limitation that hinders such observation? The authors might want to discuss this point in the manuscript.

We believe this comment originates from our unclear wording in the original version of the manuscript. In ensemble experiments, the core@shell synthesis is typically conducted at 90 °C to increase its rate. However, this is a classical thermally-driven process for which the reaction rate follows an Arrhenius (exponential) type of relationship with the operating temperature. As such, no threshold is expected. To avoid any confusion, we have rephrased the following part in the main text from:

For this purpose, we choose a synthesis of Au@CeO₂ core@shell nanoparticles that leads to a CeO₂ shell thickness of ~10 nm over 1 hour heating at 90 °C, but is kinetically inhibited at room temperature²². In a typical synthesis, a Ce³⁺ salt and ethylenediaminetetraacetic acid (EDTA) are added to gold nanoparticles stabilized by cetyltrimethylammonium bromide (CTAB) in water. Upon heating to 90 °C, Ce³⁺-EDTA complexes are hydrolyzed and Ce³⁺ ions are oxidized by the dissolved O₂ at the surface of gold nanoparticles, which act as nucleation sites for the formation of CeO₂.

to:

For this purpose, we choose a synthesis of Au@CeO₂ core@shell nanoparticles that leads to a CeO₂ shell thickness of ~10 nm over 1 hour heating at 90 °C, but is too slow at room temperature to lead to any significant shell growth²². In a typical synthesis, a Ce³⁺ salt and ethylenediaminetetraacetic acid (EDTA) are added to gold nanoparticles stabilized by cetyltrimethylammonium bromide (CTAB) in water. At higher temperatures, Ce³⁺-EDTA complexes are more easily hydrolyzed and Ce³⁺ ions are oxidized by the dissolved O₂ at the surface of gold nanoparticles, which act as nucleation sites for the formation of CeO₂.

We have also rephrased the following part in the supplementary section S1:

The Au@CeO₂ core@shell nanoparticle synthesis is kinetically inhibited at room temperature, but is activated at 90 °C.

to:

The Au@CeO₂ core@shell nanoparticle synthesis is slow at room temperature, while leading to a CeO₂ shell thickness of 10 nm upon heating to 90 °C for 1 hour.

5) In Fig. 2e the authors show that the scattering signal is independent of polarization. However, they never discuss how sensitive is their scattering signal to the shell thickness. In fig. S6 they show the result of numerical calculations but what about the experimental result, which has a broader plasmon linewidth and a noisier signal? If the grown shell was differing by only a few nm, would they be able to distinguish it in the DF signal? What about the polarization dependent measurements for the Au nanorods? Connecting to my first point, TEM images might clarify this aspect. In particular, Au nanorods would be a better test because the asymmetry should be maximum if other effects (e.g. near field, hot carriers) were to play any role.

With our experimental procedure we can determine the plasmon resonance position of our nanoparticles with a typical accuracy of ± 0.01 eV, which corresponds to a shell thickness uncertainty of less than 1 nm (see also our reply to comment #7 of reviewer #1). This is a much better accuracy than we could hope to attain using TEM on our quartz substrates (see also our reply to your first major point). We have added the following underlined sentence in the main text:

These spectral changes are consistent with the growth of a ceria shell of ~15 nm in thickness around the irradiated Au nanosphere (Supplementary

S6). With our experimental procedure we can determine the plasmon resonance position of our nanoparticles with a typical accuracy of ± 0.01 eV, which corresponds to a ceria shell thickness uncertainty of less than 1 nm (Supplementary S6).

It would have been interesting to study the isotropic nature of the ceria shell on the gold nanorods, as suggested by the reviewer. However, since the transverse dimension of the nanorods is only ~ 25 nm, the scattering cross-section in the transverse direction is too small to be detected experimentally with our setup. As such, we cannot provide polarization dependent experiments with the nanorods. We have added the underlined text to the manuscript (see also our reply to comment #9 of reviewer #1):

In Fig. 4b, we plot the longitudinal LSPR of the Au NR measured before and after irradiation. Due to the narrow width of the nanorod (~ 25 nm), the transverse plasmon resonance, which is expected at ~ 500 nm, is not visible in our measured scattering spectra (see Supplementary S13).

Regarding the use of TEM, please see our reply to your comment #1.

Minor Points

I feel that the data repartition between the main text and the supplementary often does not reflect the really important points that the authors are trying to make in the manuscript.

For example, in Fig. 1 most of the space is taken by the optical set-up description and only a minor part is devoted to the data/TEM/EDX. Although in this work the simultaneous measurement of diverse optical signals is important, I find that the optical set-ups are not novel and could be very well described in detail in supplementary without loss of understanding from the reader. Instead, Fig. 1 has no clear sketch that explain the data shown in Fig1c-d and more broadly what the authors are doing. It is true that in the experimental set-up drawings (Fig1a-b) the authors have added a shell around an Au nanoparticle under laser irradiation but the detail is practically invisible for the reader. In fact, the attention is entirely focused on the large optical set-up schematic. I thus suggest that they zoom in onto the sample schematic to clearly show the nanoparticle without/with shell without/with irradiation, respectively. They can simply use some labels (e.g. Dark Field, PL) under such zoomed view to explain which optical signals they monitor, moving the detailed optical path to SI. This schematic would then make clear the data that are currently present in Fig.1 as well as the overall concept presented in this work.

Thank you for your suggestion. We have modified Figure 1 following some of your points. We would still prefer to keep the setup schematics in the Figure, to appeal to a broad range of readers that may not be too familiar with the different microscope configurations.

Another example is Fig. 3. Most of the space is devoted to the analysis of the PL signal while very little space is given to the final result (panel d). Although I totally understand that the authors are

using a novel approach to analyze the signal and thus want to clarify their process to the reader, I wonder whether the technical details might take less space in the manuscript figure in favor of fewer, clearer experimental results. In this respect, the long and complex time traces of PL could go to the SI while only a few time profiles (currently shown as tiny sketches in Fig. 3) could be shown in the manuscript with much greater clarity for the reader.

We have followed the reviewer's advice: please see the new versions of Figure 3, its caption, and its description in the main text. The full dataset of the PL time traces is now added to the discussion in the Supplementary section S11.

Overall, these are just suggestions. Yet, I invite the authors to think carefully how their figures could be improved/restructured to avoid diluting their major experimental results among the description of smaller technical details that could be instead reported in the supplementary information.

Thank you for thinking deeply about how to improve our manuscript!

Reviewer #3 (Remarks to the Author):

The reviewed manuscript presents a comprehensive experimental analysis of light-driven nanoparticle growth. Specifically, the authors investigate the growth of dielectric and semiconductor shells on metallic nanoparticles induced by the localized plasmon heating. Using dark field microscopy, the authors are able to address and characterize this phenomenon at the single particle level, being able, at the same time, to temporally follow the shell growth. Thanks to that, the authors present very clean and clear results that allow them to draw solid conclusions. In my opinion, the paper is very interesting for the broad community of nanophotonics, timely, and well-written, and thus merits publication.

Thank you very much for your kind words.

That said, I do have a few comments and suggestions that, once addressed, I believe can help to improve the manuscript:

- At the very end of page 3, the authors mention “inelastic scattering.” Although later in the paper the meaning of this becomes clear, I think the authors should expand a bit more on what they mean by that on page 3.

We have changed the following line:

We achieve this goal by exciting the plasmon resonance of individual colloidal Au nanoparticles in a tailored reactive chemical environment, and by simultaneously following the reaction in situ using inelastic scattering spectroscopy.

to:

We achieve this goal by exciting the plasmon resonance of individual colloidal Au nanoparticles in a tailored reactive chemical environment, and by simultaneously following the reaction in situ and in real time by measuring the photoluminescence (PL) of the growing core@shell nanoparticle.

- On page 8 and Supplementary S9, the authors explain that they obtain an effective thermal conductivity by taking the average of the water and quartz thermal conductivities. However, it is not clear to me that this is justified. I believe that most of the surface of the nanoparticle should be covered by water, furthermore, the part that is in contact with the substrate should actually be experiencing a contact thermal resistance that may be very different from that predicted by the bulk thermal conductance, as shown in recent papers [1]. I recommend the authors to elaborate more on how the accuracy on the estimation of the thermal conductance can impact their results.

[1] P. Zolotavin, A. Alabastri, P. Nordlander and D. Natelson, Plasmonic Heating in Au Nanowires at Low Temperatures: The Role of Thermal Boundary Resistance, *ACS Nano*, 10(7), pp. 6972-6979, (2016).

Indeed, in our temperature calculations, we use an effective thermal conductivity that is calculated as the mean of thermal conductivities of quartz and water. Previously, such an approach has been vetted according to Setoura *et al.*¹⁰, where they use a combination of numerical and experimental results to find that, if the thermal conductivities of the substrate and the surrounding medium are of the same order of magnitude, a mean thermal conductivity can be used to calculate the nanoparticle surface temperature, even for nanospheres on a substrate.

Regarding the contact thermal resistance and the report from Zolotavin *et al.*, the interface resistance only becomes significant for substrates that are kept at cryogenic temperatures of 5 - 80 K and if the temperature increase due to photothermal heating is below 50 K. This is highlighted in the following text from the manuscript by Zolotavin *et al.* (*ACS Nano*, 10(7), pp. 6972-6979, (2016)): “*The effects of thermal boundary resistance on heat dissipation from metal nanostructures at room temperature is very small and is not usually included in modeling*”. Our experiments are performed at room temperature and as such the interfacial resistance can be safely ignored.

- On page 11, the authors claim that the change on photoluminescence can be assumed to be proportional to the change on scattering cross-section. However, does the temperature change impact these two quantities in the same way?

This is an interesting point. The understanding of the origin of photoluminescence (PL) in plasmonic nanostructures is still incomplete. However, it is commonly accepted that the spectral shape of photoluminescence closely resembles the scattering cross-section of the nanoparticle. In recent experiments, Link *et al.*¹⁷ and Shen *et al.*¹⁸ demonstrate that the photoluminescence of Au nanoparticles can be approximated as:

$$PL = EDOS \times PDOS$$

where, EDOS corresponds to the electronic density of states and PDOS corresponds to the photonic density of states, which is equal to the scattering spectrum of the nanoparticle. Given the relatively narrow temperature range used in our studies (~200 K for experiments below bubble formation) we can safely assume that the EDOS does not significantly change with temperature and that PL and scattering spectra will therefore spectrally overlap. This seems to be confirmed by the good correlation between the measured PL and LSPR maxima shown in our Fig. 3c.

Due to the limited understanding on the origin of the PL in metal nanostructures, we cannot comment further on the temperature effects on our PL measurements. While these studies may be out of the scope of this manuscript, it is definitely an interesting area to explore in the future.

- The authors show that the growth of the different shells is independent of the polarization of light. This is consistent with the fact that the temperature in the nanoparticle is rather uniform due to the high thermal conductance of metals. Although an isotropic growth is probably very

useful in many cases, sometimes a non-homogeneous growth may be desired. Could the authors envision any potential approach leading to a non-homogenous growth? Perhaps using ultrafast illumination?

Photothermal heating in metals will result in fast temperature equilibration as mentioned, unless macroscopic structures are used instead of nanoparticles. Even in the case of femtosecond pulsed illuminations, the temperature equilibration within the metal nanostructures yields a homogeneous surface temperature. The growth of anisotropic hierarchical nanostructures could however be obtained by utilizing other plasmon-decay mechanisms, such as near field enhancements and plasmonic hot charge carrier ejection. Both are directions that we are currently exploring in our lab.

REFERENCES

1. Thermal Microscopy Techniques. in *Thermoplasmonics: Heating Metal Nanoparticles Using Light* (ed. Baffou, G.) 101–142 (Cambridge University Press, 2017). doi:DOI: 10.1017/9781108289801.006
2. Carattino, A., Caldarola, M. & Orrit, M. Gold Nanoparticles as Absolute Nanothermometers. *Nano Lett.* **18**, 874–880 (2018).
3. Cai, Y.-Y. *et al.* Anti-Stokes Emission from Hot Carriers in Gold Nanorods. *Nano Lett.* **19**, 1067–1073 (2019).
4. Baffou, G. *et al.* Thermal Imaging of Nanostructures by Quantitative Optical Phase Analysis. *ACS Nano* **6**, 2452–2458 (2012).
5. Baffou, G. & Quidant, R. Thermo-plasmonics: Using metallic nanostructures as nano-sources of heat. *Laser and Photonics Reviews* **7**, 171–187 (2013).
6. Rupp, J. L. M., Scherrer, B., Harvey, A. S. & Gauckler, L. J. Crystallization and Grain Growth Kinetics for Precipitation-Based Ceramics: A Case Study on Amorphous Ceria Thin Films from Spray Pyrolysis. *Adv. Funct. Mater.* **19**, (2009).
7. Grosse, V., Bechstein, R., Schmidl, F. & Seidel, P. Conductivity and dielectric properties of thin amorphous cerium dioxide films. *J. Phys. D. Appl. Phys.* **40**, 1146–1149 (2007).
8. Li, B. *et al.* (Gold Core)@(Ceria Shell) Nanostructures for Plasmon-Enhanced Catalytic Reactions under Visible Light. *ACS Nano* **8**, 8152–8162 (2014).
9. Baffou, G. *et al.* Photoinduced heating of nanoparticle arrays. *ACS Nano* **7**, 6478–6488 (2013).
10. Setoura, K., Okada, Y., Werner, D. & Hashimoto, S. Observation of Nanoscale Cooling Effects by Substrates and the Surrounding Media for Single Gold Nanoparticles under CW-Laser Illumination. *ACS Nano* **7**, 7874–7885 (2013).
11. Aibara, I., Chikazawa, J., Uwada, T. & Hashimoto, S. Localized Phase Separation of Thermoresponsive Polymers Induced by Plasmonic Heating. *J. Phys. Chem. C* **121**, 22496–22507 (2017).
12. Jack, C. *et al.* Spatial control of chemical processes on nanostructures through nano-localized water heating. *Nat. Commun.* **7**, 10946 (2016).
13. Zeng, Z.-C., Wang, H., Johns, P., Hartland, G. V & Schultz, Z. D. Photothermal Microscopy of Coupled Nanostructures and the Impact of Nanoscale Heating in Surface-Enhanced Raman Spectroscopy. *J. Phys. Chem. C* **121**, 11623–11631 (2017).
14. Setoura, K., Okada, Y. & Hashimoto, S. CW-laser-induced morphological changes of a single gold nanoparticle on glass: observation of surface evaporation. *Phys. Chem. Chem. Phys.* **16**, 26938–26945 (2014).
15. Setoura, K., Ito, S. & Miyasaka, H. Stationary bubble formation and Marangoni convection induced by CW laser heating of a single gold nanoparticle. *Nanoscale* **9**, 719–730 (2017).
16. Thermodynamics of Metal Nanoparticles. in *Thermoplasmonics: Heating Metal Nanoparticles Using Light* (ed. Baffou, G.) 36–80 (Cambridge University Press, 2017). doi:DOI: 10.1017/9781108289801.004
17. Cai, Y.-Y. *et al.* Photoluminescence of Gold Nanorods: Purcell Effect Enhanced Emission from Hot Carriers. *ACS Nano* **12**, 976–985 (2018).

18. Hu, H., Duan, H., Yang, J. K. W. & Shen, Z. X. Plasmon-Modulated Photoluminescence of Individual Gold Nanostructures. *ACS Nano* **6**, 10147–10155 (2012).

REVIEWERS' COMMENTS:

Reviewer #1 (Remarks to the Author):

The authors have addressed in detail the comments of all reviewers. While I would have liked to see additional experiments, the responses of the authors made it clear that the additional experiments might not be technically feasible with their setup and provided very detailed explanations. The work has been carried out well and has potential to become very impactful in the field. Therefore, I recommend acceptance.

Reviewer #2 (Remarks to the Author):

The authors have addressed all of my concerns. I can thus recommend publication in Nat.Comm.

Reviewer #3 (Remarks to the Author):

The authors have clearly addressed all my comments in their revised version of the manuscript, so I recommend publication of the current version of the paper.